# ON THE IMPORTANCE OF FIRTH BIAS REDUCTION IN FEW-SHOT CLASSIFICATION

**Saba Ghaffari**[*]     **Ehsan Saleh**[*]     **David A. Forsyth**     **Yu-Xiong Wang**
Department of Computer Science, University of Illinois Urbana-Champaign
`{sabag2, ehsans2, daf, yxw}@illinois.edu`

## ABSTRACT

Learning accurate classifiers for novel categories from very few examples, known as few-shot image classification, is a challenging task in statistical machine learning and computer vision. The performance in few-shot classification suffers from the bias in the estimation of classifier parameters; however, an effective underlying bias reduction technique that could alleviate this issue in training few-shot classifiers has been overlooked. In this work, we demonstrate the effectiveness of Firth bias reduction in few-shot classification. Theoretically, Firth bias reduction removes the $O(N^{-1})$ first order term from the small-sample bias of the Maximum Likelihood Estimator. Here we show that the general Firth bias reduction technique simplifies to encouraging uniform class assignment probabilities for multinomial logistic classification, and almost has the same effect in cosine classifiers. We derive an easy-to-implement optimization objective for Firth penalized multinomial logistic and cosine classifiers, which is equivalent to penalizing the cross-entropy loss with a KL-divergence between the uniform label distribution and the predictions. Then, we empirically evaluate that it is *consistently effective across the board* for few-shot image classification, regardless of (1) the feature representations from different backbones, (2) the number of samples per class, and (3) the number of classes. Furthermore, we demonstrate the effectiveness of Firth bias reduction on cross-domain and imbalanced data settings. Our implementation is available at https://github.com/ehsansaleh/firth_bias_reduction.

## 1 INTRODUCTION

Few-shot image classification is the practice of learning accurate classifiers using a small number of labeled samples (Fei-Fei et al., 2006; Vinyals et al., 2016; Wang and Hebert, 2016; Finn et al., 2017; Snell et al., 2017; Wang et al., 2020). It has a wide range of applications from face and gesture recognition (Pfister et al., 2014) to visual navigation in robotics (Finn et al., 2017). Essentially, modern few-shot classification methods can be viewed as a combination of learning (1) a strong feature representation through a backbone network (e.g., Verma et al. (2019); Gidaris et al. (2019); Tian et al. (2020)), and (2) an accurate small-sample classifier (e.g., Wang and Hebert (2016); Chen et al. (2019)). Therefore, two key questions arise in few-shot image classification: (1) *How can we obtain a strong feature representation?* and (2) *How can we train accurate classifiers using a small number of samples?* There have been many existing methods addressing the former question using a host of different techniques (Snell et al., 2017; Rusu et al., 2019; Verma et al., 2019; Mangla et al., 2020). Here we focus on the less-explored second question. For our purposes, we will use standard feature representations and methods for training the backbone model for few-shot classification. That still leaves us with a severe classifier problem than most people realize.

There is a substantial difficulty with training a classifier using a small number of samples. In particular, with very few samples, standard classification machinery is biased. In other words, although the Maximum Likelihood Estimators (MLEs) are statistically consistent and asymptotically normal (Fahrmeir and Kaufmann, 1985), it is well-established that **MLEs are biased** for a small number of $N$ samples, with bias of $O(N^{-1})$ (Cox and Snell, 1968; Box, 1971; Whitehead, 1986; Firth, 1993). Since common logistic regression models are a type of MLEs, they are also *biased* (Schaefer,

---

[*]Both authors contributed equally and were ordered alphabetically.

1983; Cordeiro and McCullagh, 1991; Firth, 1993; Steyerberg et al., 1999). Such biases increase the error rate of the few-shot trained classifiers, and so are important in few-shot learning.

In fact, there is a standard solution for bias prevention by modifying the ordinary MLEs – known as **Firth's Penalized Maximum Likelihood Estimator (PMLE)** (Firth, 1993). In the case of the exponential family of distributions, Firth has a simplified form that penalizes the likelihood by Jeffrey's invariant prior (Firth, 1993; Poirier, 1994), which is proportional to the determinant of the Fisher Information Matrix $F$. For logistic and cosine classifiers (Chen et al., 2019) which are widely used in few-shot classification, since they belong to the exponential family, Firth bias reduction can be further cast as adding a log-determinant penalty ($\log(\det(F))$) to the cross-entropy loss. While such standard strategies can control the bias in classifiers trained with very few samples, they have not been utilized in few-shot image classification tasks.

In this paper, we show that using Firth bias reduction produces *reliable improvements in a wide range of circumstances*. We achieve this by deriving a simplified yet effective Firth formulation that penalizes the Kullback–Leibler (KL) divergence between the uniform distribution of classes and the predictions, for both *multinomial* logistic regression models and cosine classifiers. Note that common regularization techniques (such as L2-regularization and label smoothing (Szegedy et al., 2016)) cannot reduce the estimation bias of *classifier weights in small-sample regimes* (Liu et al., 2020) as the Firth penalty does; these regularization techniques are mainly used to control model complexity of deep neural networks for training *feature extractor backbones in large-sample regimes*.

More concretely, our results indicate that the improvements produced by Firth bias reduction for few-shot image classification tasks are *consistent* across the board (1) on a wide range of feature representations, (2) with both balanced and imbalanced data, (3) for both single-layer and multi-layer classifiers, (4) for both logistic and cosine classifiers, and (5) over multiple datasets and problems. Importantly, we found Firth bias reduction to consistently yield *statistically significant and positive improvements*, and we did not observe any performance penalty for utilizing it. Such improvements are on the order of 0.5-2.5% and up to 3% in challenging tasks with large number of classes.

**Our main contributions** include (1) deriving a *generalized* expression for Firth bias reduction in few-shot *multinomial* logistic regression models, and providing geometrical insight into its effect on the classification probability space; (2) evaluating the efficacy of the Firth penalized multinomial logistic model in few-shot scenarios, with both balanced and imbalanced data distributions; (3) showing that Firth bias reduction can be extended beyond typical logistic models, and can be successfully adopted in cosine classifiers; and (4) providing an empirical comparison of Firth bias reduction with common regularizers such as L2 and label smoothing.

## 2 BACKGROUND

**Mathematical Notations:** In this work, we assume a multinomial logistic regression model for the classifier, with a total of $J + 1$ classes $\{0, 1, 2, \cdots, J\}$. The logistic regression weights for class $j$ is denoted as $\beta_j$ ($1 \leq j \leq J$). The class $j = 0$ is the reference class with zero logistic regression weights. We assume to have a total number of $N$ samples $\mathcal{D} = \{(x_1, y_1), \cdots, (x_N, y_N)\}$. The $i^{\text{th}}$ target $\mathbf{y}_i$ is the one-hot encoding of the $i^{\text{th}}$ label. The assignment probability of the $i^{\text{th}}$ sample to class $j$ is denoted as $\mathrm{P}_{i,j}$:

$$\mathrm{P}_{i,j} := \Pr(y_i = j | x_i) = \frac{e^{\beta_j^{\mathsf{T}} x_i}}{1 + \sum_{j'=1}^{J} e^{\beta_{j'}^{\mathsf{T}} x_i}}. \tag{1}$$

The likelihood of the sample set $\mathcal{D}$ given the weights $\beta$ is denoted as $\Pr(y|x; \beta)$:

$$\Pr(y|x; \beta) = \prod_{i=1}^{N} \sum_{j=1}^{J} \mathbf{1}[y_i = j] \cdot \mathrm{P}_{i,j}, \tag{2}$$

where $\mathbf{1}[\cdot]$ denotes the binary indicator function. Therefore, the logistic log-likelihood function $\mathcal{L}_{\text{logistic}}$ is defined as

$$\mathcal{L}_{\text{logistic}} := \sum_{i=1}^{N} \sum_{j=0}^{J} \mathbf{1}[y_i = j] \cdot \log(\mathrm{P}_{i,j}). \tag{3}$$

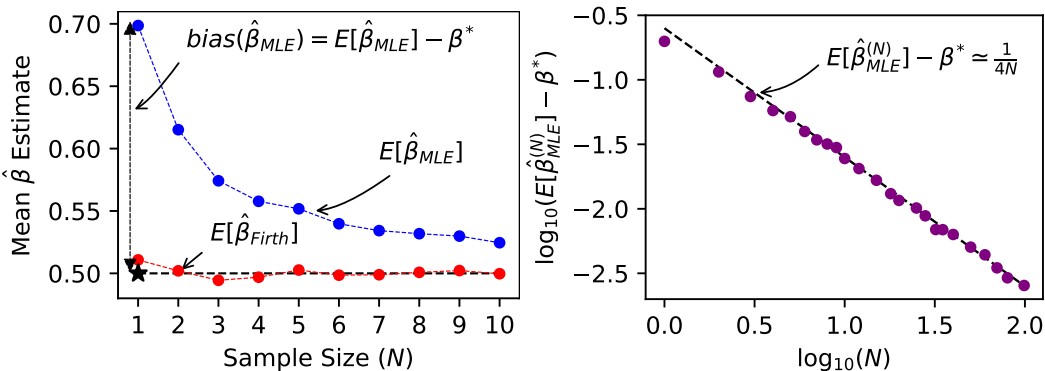

Figure 1: **The MLE bias and Firth bias reduction visualized in a geometric experiment**. Here, the task is to estimate the coin flip probability based upon the number of tosses until the first head shows up. **(Left)** The average MLE (shown in blue) and the Firth bias-reduced estimator (shown in red) against increasing number of samples. The true generative parameter $\beta^* = 0.5$ is annotated with a star, and the MLE bias is also visualized. The Firth bias-reduced estimator removes the leading $O(N^{-1})$ term from the MLE bias even with small $N$. **(Right)** To show that the MLE bias is asymptotically of $O(N^{-1})$, MLE bias is plotted against the sample size $N$ in the log-log scale.

The Fisher Information Matrix (FIM) is defined as the Hessian of the negative log-likelihood function $\mathcal{L}_{\text{logistic}}$:

$$F := -\text{Hess}_\beta(\mathcal{L}_{\text{logistic}}) = \mathbb{E}_y[\nabla_\beta \mathcal{L}_{\text{logistic}} \cdot \nabla_\beta \mathcal{L}_{\text{logistic}}^\mathsf{T}]. \tag{4}$$

The Maximum Likelihood Estimator (MLE) for logistic regression is defined as

$$\hat{\beta}_{\text{MLE}} := \arg\max_\beta \Pr(y|x;\beta). \tag{5}$$

The dimension of the feature space is denoted as $d$ in the derivations. $U_{[a,b]}$ denotes the (discrete) uniform distribution in the $[a, b]$ interval. The cross-entropy and the KL-divergence of distributions $p, q$ are defined as

$$\text{CE}(p\|q) := -\sum_x p(x) \cdot \log(q(x)), \qquad D_{\text{KL}}(p\|q) := \sum_x p(x) \cdot \log\left(\frac{p(x)}{q(x)}\right). \tag{6}$$

Table A2 in the Appendix summarizes these notations.

**Small-Sample Bias of MLE:** Assume $\beta^*$ is the true generative parameter. When the sample size $N$ is small, it is shown that the MLE bias $b(\hat{\beta}_{\text{MLE}}) := \mathbb{E}[\hat{\beta}_{\text{MLE}} - \beta^*]$ is non-zero and of $O(N^{-1})$ (Cox and Snell, 1968). Therefore, while MLE is unbiased as $N \to \infty$, it is inaccurate for few-shot learning.

**Firth Bias Reduction for MLE:** Firth's PMLE (Firth, 1993) is a modification to the ordinary MLE, which removes the $O(N^{-1})$ term from the small-sample bias. In particular, **Firth has a simplified form for the exponential family**. When $\Pr(y|x;\beta)$ belongs to the exponential family of distributions, the effect is to penalize the likelihood by Jeffrey's invariant prior (Poirier, 1994), which is proportional to the determinant of $F$ ($\det(F)$). Logistic and cosine classifiers (Chen et al., 2019) – the widely-used classification models in few-shot learning – belong to the exponential family; so for them, Firth bias reduction simplifies to adding a penalty ($\log(\det(F))$) to the cross-entropy loss. In what follows, we will derive a further simplified yet effective Firth formulation for logistic and cosine classifiers, which is computationally more efficient, deals with the case when $\det(F) = 0$, and generalizes to multinomial distributions.

**MLE vs. Firth's PMLE for a Simple Case:** To demonstrate the extent of the MLE bias and how Firth's PMLE removes the leading $O(N^{-1})$ bias term, we simulated data from the geometric distribution with probability of success $\beta = 0.5$ as its only parameter. The geometric experiment was chosen since a closed-form solution for the MLE and Firth's PMLE can be derived. Given the samples $(y_1, \cdots, y_N)$ from the geometric distribution, the sample mean is $\bar{y} = \frac{1}{N}\sum y_i$ and we have $\hat{\beta}_{\text{MLE}} = \frac{1}{\bar{y}}$ and $\hat{\beta}_{\text{Firth}} = \frac{N-1}{N\bar{y}-1}$. Note that since the sample mean is noisy, the MLE suffers from a

noisy denominator, making it biased. Figure 1 shows that $\hat{\beta}_{\mathrm{Firth}}$ is close to the true parameter $\beta^*$ for all sample sizes, whereas $\hat{\beta}_{\mathrm{MLE}}$ has a significant bias away from $\beta^*$ for small $N$. To further validate that the MLE bias is indeed of $O(N^{-1})$, we plotted the MLE bias against the sample size in the log-log scale in Figure 1, which shows that it is closely following a line with a negative unit slope.

## 3 FIRTH BIAS REDUCTION IN LOGISTIC AND COSINE CLASSIFIERS

In logistic models, the penalized likelihood function proposed by Firth is equivalent to imposing Jeffreys' prior (Poirier, 1994) on the parameters and making a maximum a posteriori estimation. In particular, Firth bias reduction encourages models with "*large*" $F$ by multiplying the likelihood by $\det(F)$. This penalty degenerates when $\det(F) = 0$. We work in the highest dimensional subspace where $F$ has full rank, and use $\det(F|r)$ to denote the product of all $r$ non-zero eigenvalues of $F$, obtaining

$$\Pr(\beta|x,y) = \frac{1}{Z} \cdot \Pr(y|x;\beta) \cdot \det(F|r)^{1/2}, \tag{7}$$

where $Z$ is a normalization constant and $\det(F|r)^{1/2}$ is the Jeffery's prior. Taking the log of both sides yields the log-posterior as a sum of the logistic log-likelihood function and the Firth bias reduction term:

$$\mathcal{L} := \log(\Pr(\beta|x,y)) = \mathcal{L}_{\mathrm{logistic}} + \mathcal{L}_{\mathrm{Firth}}, \tag{8}$$

where we have

$$\mathcal{L}_{\mathrm{Firth}} := \frac{\lambda}{2} \log(\det(F|r)) + \mathrm{cte}. \tag{9}$$

The definition of $\mathcal{L}_{\mathrm{Firth}}$ was left ambiguous up to a constant with respect to $\beta$ to facilitate the Firth bias reduction term's interpretation and avoid the definition of similar terms. Furthermore, $\lambda$ controls for the impact of the Firth term on the outcome relative to $\mathcal{L}_{\mathrm{logistic}}$. We then apply a series of derivation steps to simplify Equation (9), which are left to Section A in the Appendix. Finally, the Firth bias reduction term can be expressed as

$$\mathcal{L}_{\mathrm{Firth}} = \lambda \cdot \frac{1}{N} \sum_{i=1}^{N} \left[ \sum_{j=0}^{J} \left( \beta_j^{\mathsf{T}} x_i - \log \sum_{j'=0}^{J} e^{\beta_{j'}^{\mathsf{T}} x_i} \right) \right] = \lambda \cdot \frac{1}{N} \sum_{i=1}^{N} \left[ \sum_{j=0}^{J} \log(\mathrm{P}_{i,j}) \right]. \tag{10}$$

For the cosine classifier, the proof that $\log(\det(F))$ simplifies to Equation (10) involves straightforward manipulation of the proof for the logistic classifier, and is left to Section B in the Appendix. The normalization of the $\beta_j$ weights in the cosine classifier turns into a pure scale term in the optimization, and for a cosine classifier, scaling of the $\beta_j$ weights does not affect predictions. Therefore, this term should be ignored, and the bias reduction term effectively becomes the same as Equation (10).

**Interpreting the Firth Bias Reduction for Logistic Models:** It is well known that Jeffery's prior shrinks the parameter estimates towards zero, which is equivalent to encouraging uniform class assignment probabilities (Firth, 1993; Bull et al., 2002). We take an alternative approach to reach the same conclusion in the following. By re-arranging Equation (10), one can see the $\sum_{j=0}^{J} \log(\mathrm{P}_{i,j})$ term as a scaled average over a uniform distribution of classes:

$$\mathcal{L}_{\mathrm{Firth}} \propto \frac{1}{N} \sum_{i=1}^{N} \left[ \sum_{j=0}^{J} \frac{1}{J+1} \cdot \log(\mathrm{P}_{i,j}) \right] = \frac{-1}{N} \sum_{i=1}^{N} \left[ \mathrm{CE}\big(\mathrm{U}_{[0,J]} \| \mathrm{P}_i\big) \right]. \tag{11}$$

Therefore, we can abuse the notation, and redefine the coefficient $\lambda$ and the constant to have

$$\mathcal{L}_{\mathrm{Firth}} = \lambda \cdot \frac{-1}{N} \sum_{i=1}^{N} D_{\mathrm{KL}}\big(\mathrm{U}_{[0,J]} \| \mathrm{P}_i\big) + \mathrm{cte}. \tag{12}$$

This means that by dropping the constants, the optimization objective $\mathcal{L}$ can be rewritten as

$$\mathcal{L} = \frac{-1}{N} \sum_{i=1}^{N} \left[ \mathrm{CE}(\mathbf{y}_i \| \mathrm{P}_i) + \lambda \cdot D_{\mathrm{KL}}\big(\mathrm{U}_{[0,J]} \| \mathrm{P}_i\big) \right]. \tag{13}$$

Since the KL-divergence and the FIM define an information geometry and a Riemanian metric on probabilistic measure spaces (Nielsen, 2020), a geometrical insight into Equation (13) is provided in Figure A5 in the Appendix as well.

**Firth Bias Reduction vs. Common Regularization:** While this insight brings Firth bias reduction closer to the common regularization techniques (e.g., L2-regularization), it is worth noting that (1) common regularization techniques mainly focus on controling model complexity, instead of reducing small-sample estimation bias; (2) Firth bias reduction operates on a much lower-dimensional *target distribution* space, unlike L2 which operates in the high-dimensional *parameter* space; (3) Firth uses the same kind of metric as the logistic loss; and (4) Firth bias reduction is dimensionally consistent like the Natural Gradients (Amari, 1998; Pascanu and Bengio, 2013), whereas L2 is not.

**Firth Bias Reduction vs. Label Smoothing:** Notice that the original form of label smoothing used for training in large-sample regimes (Szegedy et al., 2016) has the same formulation as the simplified Firth penalty term in Equation (12) *for multinomial logistic classifiers*. However, Firth bias reduction is inherently different in that it reduces the classifier estimation bias in the small-sample regimes, whereas label smoothing penalizes over-confident predictions when training deep neural networks with large amounts of samples. Generally, the Firth bias reduction term (i.e., $\log(\det(F))$) is not the same as the label smoothing penalty (i.e., $D_{\mathrm{KL}}(\mathrm{U}\|\mathrm{P}_i)$) for deep neural networks. Additional analysis and empirical comparisons are provided in Section 4.4 and Section G in the Appendix.

# 4 EXPERIMENTAL RESULTS

Here we show that for a wide range of experiments, Firth bias reduction is a reliable source of small yet useful improvements in the performance. Since we report improvements for a wide range of methods and settings, the absolute accuracy improvements were reported. These absolute improvements sometimes constitute significantly to the baseline accuracy in terms of relative importance.

**Datasets:** We perform experiments on four widely-used and publicly available benchmarks: mini-ImageNet (Vinyals et al., 2016), CIFAR-FS (Bertinetto et al., 2019), tiered-ImageNet (Ren et al., 2018), and CUB (Wah et al., 2011). Each dataset consists of non-overlapping base, validation, and novel classes. The detailed class splits are described in Section D in the Appendix. Following the standard practice (Chen et al., 2019), we train feature backbones on base classes, cross-validate bias reduction coefficients on validation classes, and train classifiers and measure test accuracy over multiple trials on novel classes.

**Implementation Details:** Details regarding the setup, implementation, statistical significance, and *reducing the effect of randomized factors* are covered in Appendix Section D.

**Baselines and Evaluation Metric:** Non-penalized classifiers are used as the baseline in all experiments to compare Firth bias reduction and L2-regularization. *Absolute* accuracy improvements over the baseline averaged across multiple trials are used as the evaluation metric in all experiments. *Relative* improvements are also shown in the Appendix, which demonstrate similar behaviors.

**Summary of Results:** Section 4.1 shows the efficacy of Firth bias reduction on standard feature backbones (ResNets with varying depth) and single-layer logistic classifiers. Section 4.2 shows and argues that Firth bias reduction outperforms L2-regularization on few-shot classification tasks. Next, we investigate the driving factor in Firth's improvement in Section 4.3, and show evidence for the efficacy of Firth's bias suppression property. We also show that Firth bias reduction can be effectively, and without any modifications, applied to imbalanced data distribution settings. Section 4.4 compares Firth bias reduction against label smoothing variants, and shows that Firth outperforms label smoothing. In Section 4.5 Firth bias reduction is applied to modern few-shot methods with advanced feature backbones (i.e., WideResNet trained with strong regularization (Mangla et al., 2020)) and cosine classifiers. Experiments with additional feature backbones (DenseNet and MobileNet (Wang et al., 2019)) are included in Appendix Section F. Finally, Section 4.6 demonstrates that Firth bias reduction produces reliable improvements over the state-of-the-art feature calibration method (Yang et al., 2021). Our collective results clearly indicate a consistent pattern of improvements over a large array of (1) feature representations, (2) datasets, (3) classification ways, (4) number of shots, (5) types of classifiers, and (6) with both balanced and imbalanced data distributions.

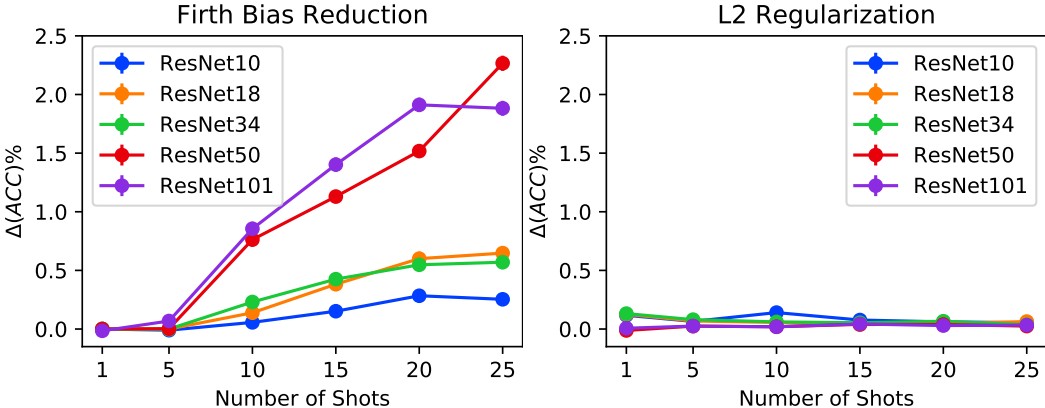

Figure 2: **Firth bias reduction produces statistically significant improvements over a wide range of backbone architectures and number of shots, whereas the common L2-regularization cannot**. Firth bias reduction (**left**) delivers a novel class accuracy improvement over the baseline classifier with an order of around 0.5-2.5%. By contrast, L2-regularization (**right**) is mostly statistically insignificant in the same setting. **16-way** logistic classification on the mini-ImageNet dataset was conducted in this experiment, with over **800 randomized and matching trials** for each combination of method, backbone, and number of samples, and the confidence intervals are covered by the blobs. Absolute and delta accuracies are provided in Table A9. The same experiments were repeated with a 3-layer classifier instead of a 1-layer classifier in Figure A7 in the Appendix and show similar results.

### 4.1 FIRTH BIAS REDUCTION IMPROVES THE ACCURACY OF LOGISTIC CLASSIFIERS

Figure 2 summarizes the results. For all the $k$-shot tasks and backbones, the average absolute test accuracy improvement is significant and it increases as $k$ increases. Also, Firth bias reduction does not hurt the 1- and 5-shot performance, with slight improvements for some backbones. Extra experiments are included in Appendix Section E.1.

### 4.2 L2-REGULARIZATION IS NOT AS EFFECTIVE AS FIRTH BIAS REDUCTION

To explore whether the performance improvement pattern for Firth bias reduction can be reproduced by common regularization techniques such as L2, the same experiments were repeated with L2-regularized logistic classifiers. For almost all the $k$-shot tasks and feature backbones, the average absolute test accuracy improvement is close to zero, as shown in Figure 2. This suggests that the bias reduction property of Firth plays a significant part in improving the classifier's performance in the few-shot regime which cannot be achieved by L2-regularization.

### 4.3 BIAS OR PRIOR?

Firth bias reduction clearly helps. It may be doing so because it is an effective way of suppressing bias. Alternatively, reinforcing the prior information might be helping. This section offers evidence that bias suppression is what is important. We consider data where the class frequencies are imbalanced. We apply a variant of Firth bias reduction that uses class prior frequencies to this data. This variant is not as successful as routine Firth bias reduction, suggesting that bias correction is what is important. One might consider replacing the uniform distribution in the Firth prior with a class probability distribution. That is, replacing $\mathcal{L}_{\text{Firth}} = -\lambda \cdot D_{\text{KL}}(\text{U}_{[0,J]} \| \text{P}_i)$ by $\mathcal{L}_{\text{Firth}} = -\lambda \cdot D_{\text{KL}}(A \| \text{P}_i)$ with $A$ being the imbalanced class distribution, when training a Firth penalized few-shot logistic classifier.

To test this, we designed two schemes to generate imbalanced datasets. These schemes differ in the imbalanced count vectors used to create the few-shot dataset in validation and novel splits. The increments in the counts per class were designed to result in two different average counts of 7.5 (Scheme 1) and 15 (Scheme 2) over the classes (experimental details are left to Section D in the Appendix). For each scheme, the experiments were carried out similar to the "balanced few-shot" case on mini-ImageNet, except that in each trial three classifiers – (1) baseline (not penalized), (2) Firth penalized (KL with the uniform prior), and (3) modified Firth penalized (KL with the imbalanced

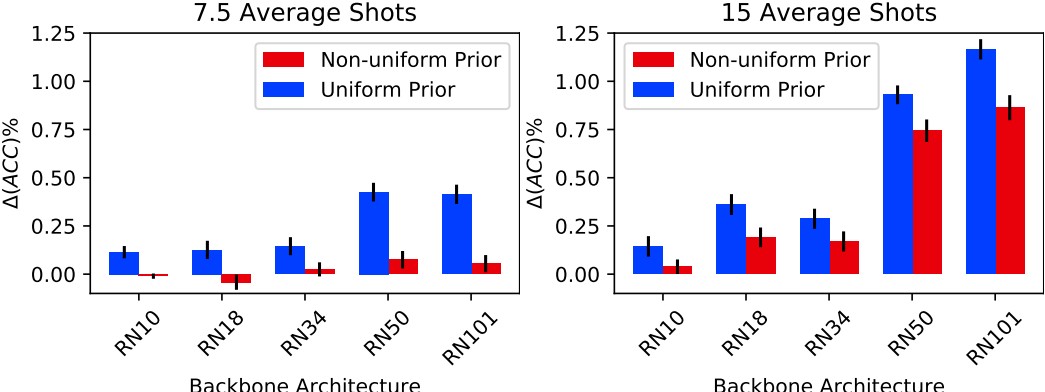

Figure 3: **Penalizing KL to the imbalanced class distribution is worse than using the Firth bias reduction.** The accuracy improvement is compared (1) when $D_{\mathrm{KL}}(U\|\mathrm{P}_i)$ is penalized (i.e., the blue bars with the uniform prior label) vs. (2) when the KL divergence to the imbalanced class distribution $A$ (i.e., $D_{\mathrm{KL}}(A\|\mathrm{P}_i)$) is penalized (i.e., the red bars with the non-uniform prior label). The left and right plots use a class distribution with an average of 7.5 and 15 shots, respectively. The range of shots vary from 2- to 16-shots in the left plot, and from 1- to 29-shots in the right plot. The difference of accuracy improvements between the uniform and the non-uniform priors is more striking for the 7.5-average-shot case, which suffers from a greater deal of estimation bias than the 15-average-shot scenario. **This shows that the improvements of Firth bias reduction are driven by its bias suppression property, and are not a mere consequence of imposing a uniform class prior.** The same experiments were repeated with a 3-layer classifier in Figure A10 in the Appendix.

prior) – were trained. In both schemes, the improvement achieved by the Firth penalized classifier is *higher* than that of the modified Firth penalized classifier over the baseline for all backbones, as shown in Figure 3. The difference between the two improvements is more substantial in Scheme 1, where the average of samples per class is less than Scheme 2. This implies that the Firth penalized classifier is indeed reducing the bias and should not be naively considered as if a prior is simply defined over the class assignment probabilities.

## 4.4 FIRTH BIAS REDUCTION IMPROVES OVER LABEL SMOOTHING

Even though the original version of label smoothing (Szegedy et al., 2016) has the same formulation as the Firth bias reduction term in Equation (13), there is a family of label smoothing techniques that aim to reduce overfitting and over-confident predictions (Pereyra et al., 2017). Namely, the confidence penalty (i.e., regularizing $D_{\mathrm{KL}}(\mathrm{P}_i\|U)$) and unigram label smoothing (i.e., regluarizing the KL-divergence with class priors) are two label smoothing variants, which are shown to outperform the original label smoothing, when training complex networks with large amounts of samples (Pereyra et al., 2017). Figure 3 shows that Firth bias reduction outperforms unigram label smoothing, and Table A4 in the Appendix shows that Firth bias reduction yields better results than applying confidence penalty and entropy regularization. This further suggests that the improvements obtained by Firth are the result of its bias reduction property.

## 4.5 FIRTH BIAS REDUCTION IMPROVES THE PERFORMANCE OF COSINE CLASSIFIERS

As cosine classifiers have recently proven to be useful for few-shot classification (Gidaris and Komodakis, 2018; Chen et al., 2019), we investigated the impact of Firth bias reduction on them. The difference between a logistic and cosine classifier parameter space is that the parameter space of the cosine classifier is constrained to a unit sphere, therefore it is likely for the Firth bias reduction to improve cosine classifier performance as well. To test this, we based our experiments on the implementation of the S2M2$_R$ method (Mangla et al., 2020), and used their pre-trained WideResNet-28-10 (Zagoruyko and Komodakis, 2016) on the base classes as a strong feature extractor. We trained Firth penalized and non-penalized cosine classifiers on the mini-ImageNet, tiered-ImageNet, and CIFAR-FS datasets. Few-shot classifiers were trained for varying number of classes, depending on

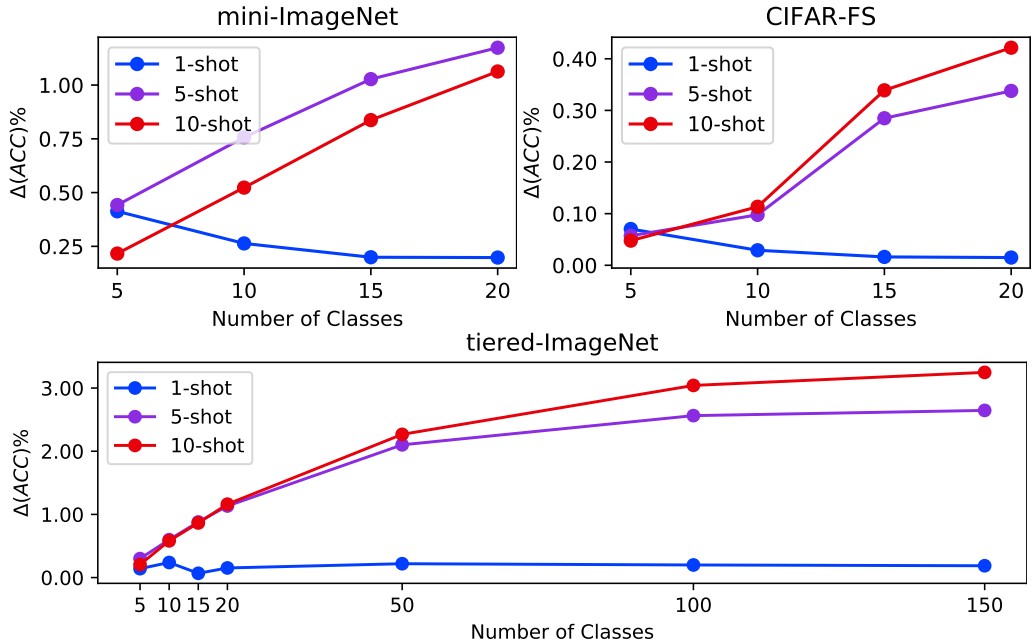

Figure 4: **Firth bias reduction consistently improves the accuracy of the cosine classifier (a better few-shot classifier) in the presence of strong backbone features, regardless of (1) the type of dataset, and (2) the number of classes in few-shot classification.** For each dataset, an advanced backbone architecture (WideResNet-28-10) with strong mixup regularization and self-supervision was trained. The error bars are almost non-existent (i.e., less than $0.02\%$), since over 10,000 trials with **matching randomization factors** were performed for each point. **Firth bias reduction improvements are always positive, and it never hurts to use the Firth bias reduction technique.** Absolute and delta accuracies are provided in Tables A6, A7, and A8 in the Appendix.

the dataset, and varying number of samples per class. We followed the same scheme as Mangla et al. (2020) to generate the $k$-shot support and query datasets for training and testing, respectively.

As shown in Figure 4, in all datasets, Firth bias reduction improves the performance of 5- and 10-shot tasks in a monotonically increasing fashion with respect to the number of classes. This behavior is well represented for tiered-ImageNet, as increasing 5-way to 150-way classification results in more than 2.5% improvement of the average test accuracy. In the 1-shot task, the improvement by Firth bias reduction is monotonically decreasing with the number of classes in all the datasets except tiered-ImageNet, but it does not hurt the performance. The improvement almost falls within (0.1%-0.5%), (0-0.1%), and (0-0.2%), for mini-ImageNet, CIFAR-FS, and tiered-ImageNet, respectively.

This experiment suggests that not only is Firth bias reduction effective for cosine classifiers, but it also could help with the performance even when strong features are used. This is well justified as Firth bias reduction targets the bias introduced in the classifier parameters not the features.

## 4.6 COMPARISON WITH ADDITIONAL STATE-OF-THE-ART METHOD

We further applied the Firth bias reduction term to the recent state-of-the-art method in few-shot classification (Yang et al., 2021), called Distribution Calibration (DC). DC computes the closest base classes to each sample in the feature space, and samples artificial examples from a Gaussian distribution centered around the mean of the closest base class features. To make the setup challenging, we consider data where the base classes are categorically different from the novel classes: i.e., by performing *cross-dataset* few-shot classification, where the features are trained on the base classes of mini-ImageNet or tiered-ImageNet, and the novel sets are instead sampled from the CUB dataset. The results are shown in Table 1, and indicate that the Firth bias reduction improves over Yang et al. (2021). We also show the Firth bias reduction improvements on the tiered-ImageNet dataset in

| | | mini-ImageNet → CUB | | | tiered-ImageNet → CUB | | |
|---|---|---|---|---|---|---|---|
| Way | Shot | Before | After | Improvement | Before | After | Improvement |
| 10 | 1 | 37.14 | $37.40 \pm 0.13$ | $0.26 \pm 0.03$ | 64.36 | $64.52 \pm 0.16$ | $0.15 \pm 0.04$ |
| 10 | 5 | 59.77 | $60.77 \pm 0.12$ | $1.00 \pm 0.04$ | 86.23 | $86.66 \pm 0.10$ | $0.43 \pm 0.02$ |
| 15 | 1 | 30.22 | $30.37 \pm 0.09$ | $0.15 \pm 0.03$ | 57.73 | $57.74 \pm 0.13$ | $0.01 \pm 0.01$ |
| 15 | 5 | 52.73 | $53.84 \pm 0.10$ | $1.12 \pm 0.03$ | 82.16 | $83.05 \pm 0.09$ | $0.90 \pm 0.02$ |

Table 1: **The cross-domain experiments for the DC method with and without Firth bias reduction.** The columns containing the novel set accuracy obtained by DC and Firth penalized DC methods are tagged *Before* and *After*, respectively. Each setting (a combination of way, shot, and method) was tested with and without data augmentation (addition of 750 samples), and the maximum accuracy was reported. Note that the confidence intervals are much smaller for the improvement column, thanks to the random-effect matching procedure we used in this study. The *Before* confidence intervals were similar to the *After* confidence intervals, and thus not repeated due to space constraints.

Appendix Section H and Table A5. This further supports the observation that Firth bias reduction yields small but reliable improvements under different methods.

## 5 RELATED WORK

**Bias Reduction of the MLE:** A myriad of statistical work has been proposed to mitigate the small-sample bias of the MLE under different settings (Anderson and Richardson, 1979; Kenward and Roger, 1997; Bull et al., 2002; Kosmidis and Firth, 2009). Originally, the asymptotic bias of MLE was shown to be of $O(N^{-1})$, with $N$ being the sample size (Firth, 1993). To counter such an estimation bias, many approaches have existed. To name a few, (1) additive penalization terms to the main logistic loss were proposed to reduce the MLE's bias (Firth, 1993; Bull et al., 2002; Greenland and Mansournia, 2015), and (2) some methods have been proposed to directly approximate and remove such a small sample bias (Cox and Hinkley, 1979). While the latter approach may sound appealing, estimating the MLE's bias can be impractical. For instance, in few-shot scenarios, a perfect separation of the classes may be achievable, causing the logistic MLE to be unbounded (Heinze and Schemper, 2002). On the other hand, the penalization methods do not modify the estimated parameters directly, and instead gently push for a preference towards less biased estimates. Such penalization methods can be generally applicable to a vast array of models.

Theoretically, Firth's PMLE reduces the bias by removing the leading $O(N^{-1})$ term from the MLE's bias (Firth, 1993) – a property that does not exist in common regularization techniques such as L2-regularization. Furthermore, PMLE of the logistic model has been shown to have smaller variance than MLE as well (Copas, 1988; Kosmidis and Firth, 2009). Firth's PMLE has been well studied for binomial logistic regression (Firth, 1993), and applied and tested against other penalization techniques in other fields (Rainey and McCaskey, 2015; Muchlinski et al., 2016; Rahman and Sultana, 2017).

Additional related work on **few-shot image classification** was left to Appendix Section C.

## 6 CONCLUSION

We show that Firth bias reduction consistently improves the accuracy across the board in few-shot classification regardless of (1) the employed feature backbone, (2) the number of classes and samples, and (3) the dataset and problem setting. Furthermore, our experiments show that Firth bias reduction can improve the performance of the cosine classifiers, and is applicable to imbalanced and cross-domain few-shot settings without any necessary modifications. Overall, our evaluations suggest that Firth bias reduction is a useful and general bias reduction tool that has been missing in few-shot classification, and should be incorporated in few-shot classification tasks for accuracy improvements.

## 7 ACKNOWLEDGMENT

This work was supported in part by (1) the National Science Foundation's Major Research Instrumentation program (Kindratenko et al., 2020), grant number 1725729, (2) the National Science Foundation's Creating Knowledge with All-Novel-Class Computer Vision program, grant number 2106825, and (3) the Jump ARCHES endowment through the Health Care Engineering Systems Center. The majority of NSF funding contributions were designated to Ehsan Saleh and Saba Ghaffari equally in the form of computational resource allocations on high-performance computing platforms for conducting the experiments. Overall, this work consumed more than 32 CPU years and one Nvidia V-100 GPU year from the NSF-funded resource allocations in the course of its analysis. Also, this work made use of the Illinois Campus Cluster, a computing resource that is operated by the Illinois Campus Cluster Program (ICCP) in conjunction with the National Center for Supercomputing Applications (NCSA) and is supported by funds from the University of Illinois Urbana-Champaign.

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

# A FIRTH BIAS REDUCTION FOR FEW-SHOT MULTINOMIAL LOGISTIC REGRESSION

Table A2 summarizes the notations used throughout the main paper and here. Given a dataset of $N$ samples $\mathcal{D} = \{(x_1, y_1), (x_2, y_2), \cdots, (x_N, y_N)\}$, the multinomial logistic model for a total of $J + 1$ classes can be formulated as

$$\log\left[\frac{\Pr(y = j | x_i)}{\Pr(y = 0 | x_i)}\right] = \beta_j^{\mathsf{T}} x_i, \qquad j \in \{1, 2, \cdots, J\}, \tag{A14}$$

where class $j = 0$ was chosen as the reference class in the log odds ratio. In other words, w.l.o.g. we assume $\beta_0 = 0$ in this formulation. Given the decision rule in Equation (A14), we can write

$$P_{i,0} = \frac{1}{1 + \sum_{j'=1}^{J} e^{\beta_{j'}^{\mathsf{T}} x_i}}, \qquad P_{i,j} = \frac{e^{\beta_j^{\mathsf{T}} x_i}}{1 + \sum_{j'=1}^{J} e^{\beta_{j'}^{\mathsf{T}} x_i}} \qquad \forall 1 \le j \le J. \tag{A15}$$

Under this notation, the log-likelihood would be $\mathcal{L}_{\text{logistic}} = \sum_{i=1}^{N} \sum_{j=0}^{J} \mathbf{1}[y_i = j] \cdot \log(P_{i,j})$. The data matrix $X_D$ is given as

$$X_D := [x_1 \quad x_2 \quad \cdots \quad x_N]_{d \times N}. \tag{A16}$$

Also, $X := X_D \otimes I_J$ where the $\otimes$ operator denotes the Kronecker matrix product, and the $J$-dimensional identity matrix is denoted as $I_J$.

Firth (1993) has established that the bias of logistic regression can be removed by maximizing the sum of (1) the log-likelihood $\mathcal{L}_{\text{logistic}}$ and (2) the log-determinant of the Fisher Information Matrix (FIM). For our purposes, this presents some challenges: we have a few number of samples and the FIM determinant is zero. Instead, we use the product of all non-zero eigenvalues of the FIM as its "*amended determinant*". To obtain this efficiently, we need to know the specific structure of the FIM.

It is important in what follows that the FIM can be defined as the matrix product

$$F_{(dJ) \times (dJ)} = X_{(dJ) \times (NJ)}^{\mathsf{T}} \cdot M_{(NJ) \times (NJ)} \cdot X_{(NJ) \times (dJ)}, \tag{A17}$$

where $M$ is a block-diagonal matrix whose $i^{\text{th}}$ diagonal block is denoted as $M_i$. We leave the definition of $X$ and $M_i$ to the "*FIM Formulation for Logistic Regression*" subsection. Next, we focus on:

- *Determinant Amendment and Constant Dropping*: Generically, we show that

$$\log(\det(F|NJ)) = \sum_{i=1}^{N} \log(\det(M_i)) + \text{cte}, \tag{A18}$$

  where $\det(F|NJ)$ is an amended version of $\det(F)$.

- *Efficient Computation of* $\log(\det(M_i))$: Next, we show that

$$\log(\det(M_i)) = \sum_{j=0}^{J} \log(P_{i,j}). \tag{A19}$$

Combining these two points will lead us to the simplified Firth bias reduction objective:

$$\mathcal{L}_{\text{Firth}} = \lambda \cdot \frac{1}{N} \sum_{i=1}^{N} \sum_{j=0}^{J} \log(P_{i,j}). \tag{A20}$$

**Determinant Amendment and Constant Dropping:** Having $F = X^{\mathsf{T}} \cdot M \cdot X$ prompts us to utilize the SVD of $X$ as

$$X_{NJ \times dJ} = U_{NJ \times NJ} \cdot S_{NJ \times dJ} \cdot V_{dJ \times dJ}^{\mathsf{T}}. \tag{A21}$$

Therefore, the FIM can be written as $F = V \cdot (S^{\mathsf{T}} \cdot K \cdot S) \cdot V^{\mathsf{T}}$, where $K := U^{\mathsf{T}} \cdot M \cdot U$. Since $V$ and $U$ are rotation matrices, we can write $\det(F) = \det(S^{\mathsf{T}} \cdot K \cdot S)$, and

$$\det(K) = \det(M) = \prod_{i=1}^{N} \det(M_i). \tag{A22}$$

| Notation | Description |
|---|---|
| $J + 1$ | Total Number of Classes in Multinomial Logistic Regression |
| $\beta_j$ | The Logistic Regression Weights for Class $j$ ($j \in \{1, 2, \cdots, J\}$) |
| $\mathbf{P}_{i,j}$ | Classification Probability of Sample $i$ Belonging to Class $j$ |
| $\mathbf{P}_i$ | The $i^{\text{th}}$ Sample's Soft Classification Probabilities |
| $N$ | Number of Samples |
| $\mathcal{D} = \{(x_i, y_i)\}_{i=1}^{N}$ | Logistic Regression Sample Dataset |
| $\mathbf{y}_i$ | The One-Hot Encoding of the Label $y_i$ |
| $\mathbf{1}[a = b]$ | The Binary Indicator Function (i.e., 1 when $a = b$ and 0 otherwise) |
| $F$ | The Fisher Information Matrix |
| $d$ | Dimension of the Features |
| $I_J$ | The $J \times J$ Identity Matrix |
| $A \otimes B$ | The Kronecker Product of Matrix $A$ by Matrix $B$ |
| $\mathbf{1}_{r \times c}$ | The All Ones Matrix with $r$ Rows and $c$ Columns |
| $\hat{\beta}_{\text{MLE}}$ | The Maximum Likelihood Estimator (MLE) |
| $\mathcal{L}_{\text{logistic}}$ | The Logistic Log-Likelihood Function |
| $\mathcal{L}_{\text{Firth}}$ | The Firth Bias Reduction Function |
| $\lambda$ | The Firth Bias Reduction Coefficient |
| $\text{CE}(p\|q)$ | The Cross-Entropy of $p$ and $q$ |
| $D_{\text{KL}}(p\|q)$ | The KL Divergence of $p$ and $q$ |
| $\text{U}_{[0,J]}$ | The Uniform Class Assignment Probabilities |
| $\det(A|r)$ | The Amended Determinant of the Degenerate Matrix $A$ with at most $r$ Non-zero Eigenvalues (See Section A in the Appendix) |

Table A2: The mathematical notations used throughout the paper.

As we have $d > N$ for most few-shot tasks, the matrix $S$ can be viewed in the following form:

$$S_{NJ \times dJ} = \begin{bmatrix} \hat{S}_{NJ \times NJ} & 0 \end{bmatrix}, \tag{A23}$$

where $\hat{S}$ is a diagonal square matrix. Since $\hat{S}^{\mathsf{T}} = \hat{S}$, we have

$$S^{\mathsf{T}} \cdot K \cdot S = \begin{bmatrix} \left( \hat{S} \cdot K \cdot \hat{S} \right)_{NJ \times NJ} & 0 \\ 0 & 0 \end{bmatrix}. \tag{A24}$$

Equation (A24) and $\det(F) = \det(S^{\mathsf{T}} \cdot K \cdot S)$ show why $\det(F)$ is zero. For mitigation, we replace $\det(F)$ with the product of the non-zero eigenvalues of $F$, namely $\det(F|NJ)$, and call it the "*amended determinant*" of $F$. Thanks to $F = V \cdot \left( S^{\mathsf{T}} \cdot K \cdot S \right) \cdot V^{\mathsf{T}}$, even the amended determinants $\det(F|NJ)$ and $\det(S^{\mathsf{T}} \cdot K \cdot S|NJ)$ are the same:

$$\det(F|NJ) = \det(S^{\mathsf{T}} \cdot K \cdot S|NJ) = \det(\hat{S} \cdot K \cdot \hat{S}) = \det(M) \cdot \det(\hat{S}^2). \tag{A25}$$

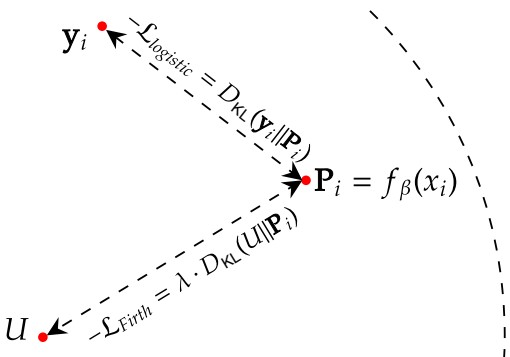

Figure A5: **Firth bias reduction is effectively the same as minimizing the KL divergence to the uniform class probabilities for logistic regression**. Here, $\mathrm{P}_i$ denotes the predicted distribution of classes for the sample $x_i$, and $U$ denotes the uniform distribution of classes. The logistic objective minimizes the KL-divergence between the true label $\mathbf{y}_i$ and $\mathrm{P}_i$, while Firth bias reduction $\mathcal{L}_{\text{Firth}}$ tries to tie $\mathrm{P}_i$ with a *KL divergence rope* to the uniform distribution over the classes.

*Could* $\det(F|NJ)$ *be zero?* The answer is "*not*" generically; $\det(M)$ is generically positive as we will show $\det(M_i) = \prod_{j=0}^{J} \mathrm{P}_{i,j}$ later in Equation (A28). Also, $\det(\hat{S}^2) > 0$ holds with probability 1 for continuous data distributions. In fact, $\det(\hat{S}^2)$ can only be zero when the data contains linearly dependant samples, which happens with zero probability for non-atomic data distributions. Therefore, we have

$$\log(\det(F|NJ)) = \sum_{i=1}^{N} \log(\det(M_i)) + \log(\det(\hat{S}^2)). \tag{A26}$$

This is the same as Equation (A18): since the $\log(\det(\hat{S}^2))$ term is independent of the model's parameters, we can treat it as an optimization constant and drop it.

**Efficient Computation of** $\log(\det(M_i))$**:** We define the soft predictions of the $i^{\text{th}}$ sample (excluding the reference class) as $\mathrm{P}_{i,1:J} := \begin{bmatrix} \mathrm{P}_{i,1} & \cdots & \mathrm{P}_{i,J} \end{bmatrix}^{\mathsf{T}}$. Given $M_i$'s definition in Equation (A31), we can write

$$M_i = \text{Diag}(\mathrm{P}_{i,1:J}) - \mathrm{P}_{i,1:J} \cdot \mathrm{P}_{i,1:J}^{\mathsf{T}}. \tag{A27}$$

Next, we use the Matrix-Determinant Lemma (Harville, 1998) to compute $\det(M_i)$:

$$\det(M_i) = \det(\text{Diag}(\mathrm{P}_{i,1:J})) \cdot (1 - \mathrm{P}_{i,1:J}^{\mathsf{T}} \cdot \text{Diag}(\mathrm{P}_{i,1:J}) \cdot \mathrm{P}_{i,1:J})$$

$$= (\prod_{j=1}^{J} \mathrm{P}_{i,j}) \cdot (1 - \mathrm{P}_{i,1:J}^{\mathsf{T}} \mathbf{1}_{J \times 1}) = (\prod_{j=1}^{J} \mathrm{P}_{i,j}) \cdot \mathrm{P}_{i,0} = \prod_{j=0}^{J} \mathrm{P}_{i,j}. \tag{A28}$$

Taking the log will give us Equation (A19).

**The FIM Formulation for Logistic Regression:** Elementary methods established that the FIM of the logistic classifier is composed of $J \times J$ block matrices, each with a dimension of $d \times d$ (where $d$ is the dimension of the features). These block matrices can be expressed as

$$F_{j,j} = \sum_{i=1}^{N} \mathrm{P}_{i,j} \cdot (1 - \mathrm{P}_{i,j}) \cdot x_i \cdot x_i^{\mathsf{T}} \qquad \forall \quad 1 \leq j \leq J,$$

$$F_{j,k} = -\sum_{i=1}^{N} \mathrm{P}_{i,j} \cdot \mathrm{P}_{i,k} \cdot x_i \cdot x_i^{\mathsf{T}} \qquad \forall \quad 1 \leq j \neq k \leq J. \tag{A29}$$

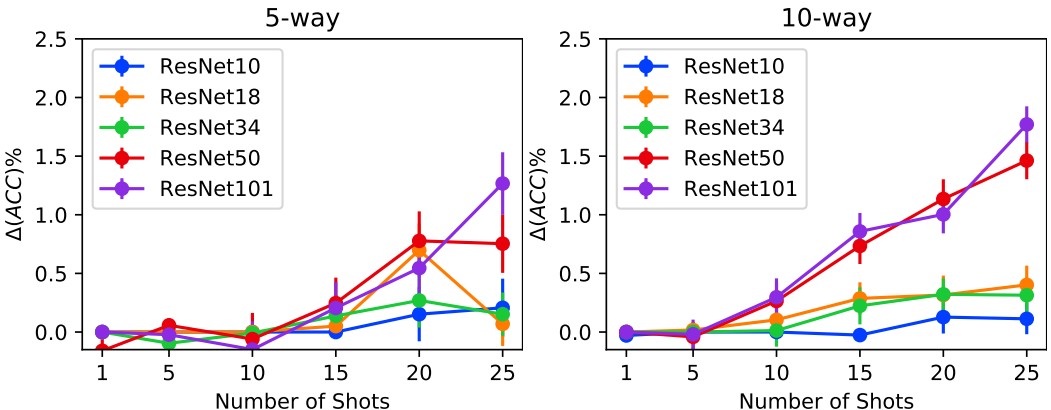

Figure A6: **Improvements of Firth-penalized logistic classifier over the baseline on mini-ImageNet with 5 and 10 number of classes.** Except for the number of classes, the experiments were performed in the same setting as in Figure 2.

This facilitates the expression of the FIM as shown in Equation (A17): We can define $X$ as the Kronecker product of (1) the data matrix $X_D$ and (2) the $J$-dimensional identity matrix $I_J$:

$$X_D := \begin{bmatrix} x_1^\mathsf{T} \\ \hline x_2^\mathsf{T} \\ \hline \vdots \\ \hline x_N^\mathsf{T} \end{bmatrix}_{N \times d} , \qquad X := X_D \otimes I_J. \tag{A30}$$

$M$ would then be a block-diagonal matrix whose $i^{\text{th}}$ block is defined as

$$\begin{aligned} (M_i)_{j,j} &= \mathrm{P}_{i,j}(1 - \mathrm{P}_{i,j}) & \forall \quad 1 \le j \le J, \\ (M_i)_{j,k} &= -\mathrm{P}_{i,j}\mathrm{P}_{i,k} & \forall \quad 1 \le j \ne k \le J. \end{aligned} \tag{A31}$$

# B    FIRTH BIAS REDUCTION FOR COSINE CLASSIFIERS

Section A derived the Firth bias reduction term for the logistic classifier as $D_{\mathrm{KL}}(\mathrm{U}\|\mathrm{P}_i)$. Here, we generalize this observation to cosine classifiers, and prove that the Firth bias reduction for cosine classifiers reduces down to the same form as the one obtained for logistic classifiers.

By defining the normalization transformation

$$T(\beta) = \left[\frac{\beta_0^\mathsf{T}}{\|\beta_0\|}, \cdots, \frac{\beta_J^\mathsf{T}}{\|\beta_J\|}\right]^T, \qquad T(x) = \left[\frac{x_1^\mathsf{T}}{\|x_1\|}, \cdots, \frac{x_N^\mathsf{T}}{\|x_N\|}\right]^T, \tag{A32}$$

we have the log-likelihood relation between the cosine classifier and the logistic model:

$$\mathcal{L}_{\text{cosine}}(\beta; x, y) = \mathcal{L}_{\text{logistic}}(T(\beta); T(x), y). \tag{A33}$$

According to Equation (4) in the main paper and the chain rule, we can write

$$F_{\text{cosine}}(\beta, x, y) = A \cdot F_{\text{logistic}}(T(\beta), T(x), y) \cdot A^\mathsf{T}, \tag{A34}$$

where $A_{d(J+1) \times d(J+1)}$ is the Jacobian matrix of $T(\beta)$ with respect to $\beta$. It can be shown that $A$ is a symmetric block-diagonal matrix, whose $j^{\text{th}}$ diagonal block is

$$A_{j,j} = \frac{1}{\|\beta_j\|}\left(I_{d \times d} - \frac{\beta_j \beta_j^\mathsf{T}}{\beta_j^\mathsf{T} \beta_j}\right). \tag{A35}$$

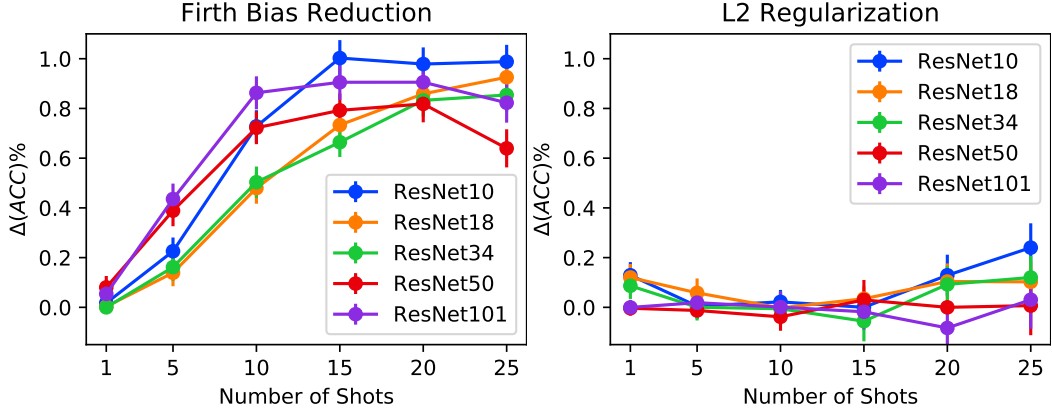

Figure A7: **Same experiment as Figure 2 in the main paper but with a 3-layer logistic classifier instead of a 1-layer logistic classifier (in the main paper)**. Again, Firth bias reduction improves the accuracy for different backbones and number of samples **(left)**, whereas L2-regularization is not effective at all **(right)**. In the very hard cases, i.e., 1- and 5-shots, there is a mild improvement by Firth and no deterioration. The error bars, representing the $95\%$ confidence intervals, are covered by the blobs.

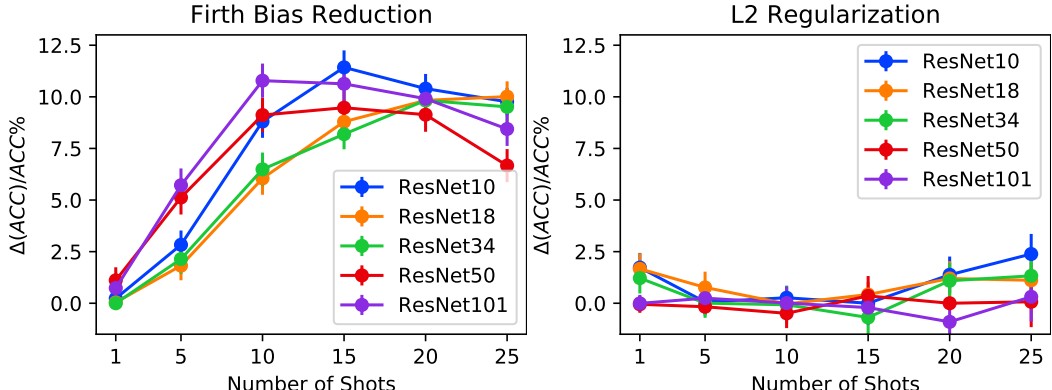

Figure A8: **Relative accuracy improvements corresponding to Figure A7 with the 3-layer logistic classifier**. The behavior of the relative accuracy improvements is similar to the absolute accuracy improvements for all the backbones in both techniques.

Therefore,

$$\log(\det(F_{\text{cosine}})) = \log(\det(F_{\text{logistic}})) + 2\log(\det(A))$$
$$= \log(\det(F_{\text{logistic}})) + 2\sum_{j=0}^{J}\log(\det(A_{j,j})). \tag{A36}$$

It is obvious that

$$A_{j,j} \cdot v = \begin{cases} 0 & v \parallel \beta_j \\ \frac{v}{\|\beta_j\|} & v \perp \beta_j. \end{cases} \tag{A37}$$

Therefore, $A_{j,j}$ can be thought as an identity-proportional matrix in the sub-space perpendicular to $\beta_j$. Therefore, its amended determinant is

$$\log(\det(A_{j,j}; d-1)) = -(d-1) \cdot \log(\|\beta_j\|). \tag{A38}$$

For the plain logistic model, the Firth penalized optimization problem is:

$$\max_{\beta} \left[ \mathcal{L}_{\text{logistic}}(\beta; x, y) + \lambda \cdot \log(\det(F_{\text{logistic}}(\beta, x, y))) \right]. \tag{A39}$$

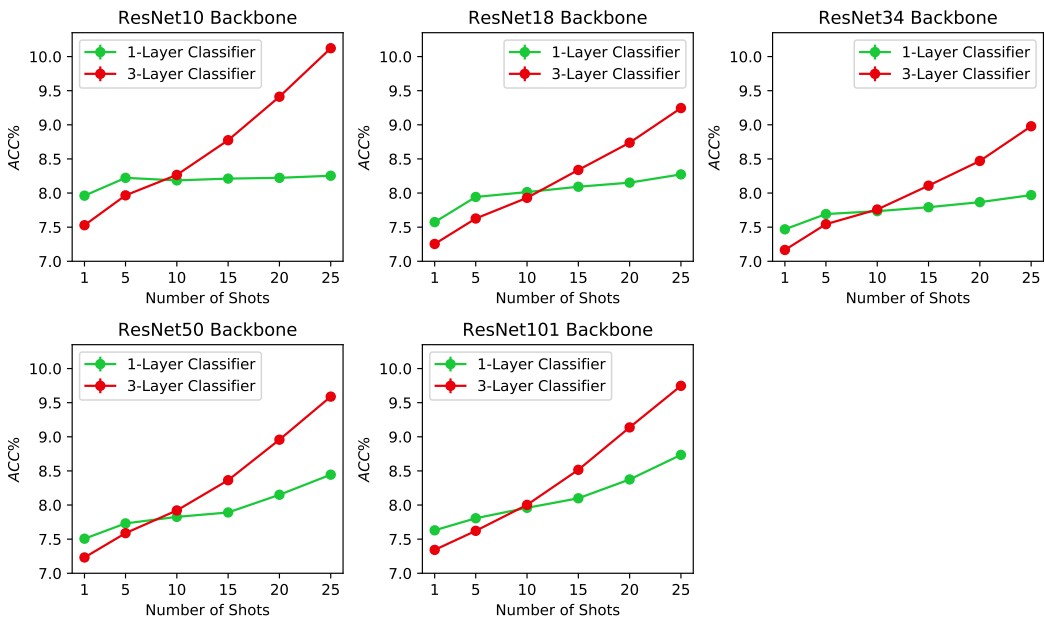

Figure A9: **Absolute test accuracy values of 16-way classification for the non-penalized 3-layer and single-layer logistic classifiers on the novel set of mini-ImageNet for different backbones and varying number of samples.**

For the cosine classifiers, the Firth penalized optimization problem is:

$$\max_{\beta} \left[ \mathcal{L}_{\text{logistic}}(T(\beta); T(x), y) + \lambda \cdot \log(\det(F_{\text{logistic}}(T(\beta), T(x), y))) - C \sum_{j=0}^{J} \log(\|\beta_j\|) \right], \quad \text{(A40)}$$

where we have $C := 2\lambda(d-1)$. By re-parameterizing $\beta = (u_\beta, s_\beta)$ with $u_\beta := T(\beta)$ and $s_\beta := \left[ \|\beta_0\|, \cdots, \|\beta_J\| \right]$, we have

$$\max_{u_\beta, s_\beta} \left[ \mathcal{L}_{\text{logistic}}(u_\beta; T(x), y) + \lambda \cdot \log(\det(F_{\text{logistic}}(u_\beta, T(x), y))) - C \sum_{j=0}^{J} \log(s_{\beta_j}) \right]. \quad \text{(A41)}$$

Since $u_\beta$ and $s_\beta$ are independent optimization parameters, this problem can be restated as

$$\max_{u_\beta} \left[ \mathcal{L}_{\text{logistic}}(u_\beta; T(x), y) + \lambda \cdot \log(\det(F_{\text{logistic}}(u_\beta, T(x), y))) \right] + \max_{s_\beta} \left[ -C \sum_{j=0}^{J} \log(s_{\beta_j}) \right].$$
$$\text{(A42)}$$

Since $s_\beta$ has no effect on the predictions of the model, it can be ignored. Therefore, we end up with the optimization problem

$$\max_{u_\beta} \left[ \mathcal{L}_{\text{logistic}}(u_\beta; T(x), y) + \lambda \cdot \log(\det(F_{\text{logistic}}(u_\beta, T(x), y))) \right]. \quad \text{(A43)}$$

Essentially, this suggests applying the same bias reduction form to cosine classifiers as the one used for logistic classifiers $\log(\det(F_{\text{logistic}}(u_\beta, T(x), y)))$ reduces down to $D_{\text{KL}}(\text{U}\|\text{P}_i)$ as proven in Section A, and $\mathcal{L}_{\text{logistic}}(u_\beta; T(x), y)$ is the cross-entropy loss with the true labels.

## C  ADDITIONAL RELATED WORK ON FEW-SHOT IMAGE CLASSIFICATION

As one of the unsolved problems in machine learning, few-shot learning (Miller et al., 2000; Fei-Fei et al., 2006) has attracted growing interest in the deep learning era (Lake et al., 2015; Santoro et al., 2016; Wang and Hebert, 2016; Vinyals et al., 2016; Snell et al., 2017; Finn et al., 2017; Hariharan and

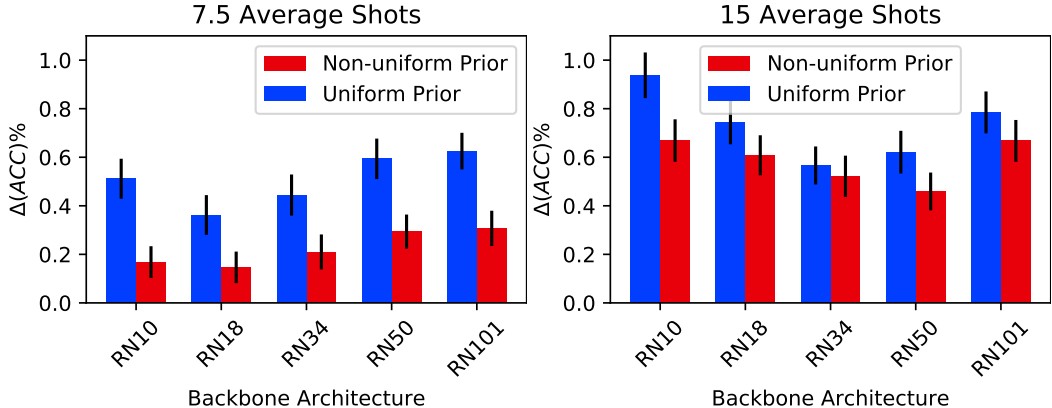

Figure A10: **Same experiment as Figure 3 in the main paper but with a 3-layer logistic classifier instead of a single-layer (in the main paper).** Again, in both schemes (7.5 and 15 average shots), the Firth penalized classifier results in more accuracy improvements than the classifier penalized with KL-divergence to the non-uniform prior. This further suggests that Firth bias reduction is indeed reducing the bias and is not simply imposing a uniform prior. Also, it reaffirms that plain Firth bias reduction can safely be used on imbalanced data.

Girshick, 2017; George et al., 2017; Triantafillou et al., 2017; Edwards and Storkey, 2017; Mishra et al., 2018; Douze et al., 2018; Wang et al., 2018; Chen et al., 2019; Dvornik et al., 2019; Allen et al., 2019; Li et al., 2019a; Yoon et al., 2019; Lifchitz et al., 2019; Li et al., 2019b; Zhang et al., 2020; Ye et al., 2020; Dhillon et al., 2020; Triantafillou et al., 2020; Tian et al., 2020; Dvornik et al., 2020; Yang et al., 2021; Phoo and Hariharan, 2021; Gui et al., 2021; Liu et al., 2021; Zhang et al., 2021). Successful generalization from few training samples requires "inductive biases" or shared knowledge from related tasks (Baxter, 1997), which is commonly acquired through transfer learning, and more recently, meta-learning (Thrun, 1998; Schmidhuber, 1987; Schmidhuber et al., 1997; Bengio et al., 2013). By explicitly "learning-to-learn" over a series of few-shot learning tasks (i.e., episodes), which are simulated from base classes, meta-learning exploits accumulated task-agnostic knowledge to few-shot learning problems of novel classes. Within this paradigm, various types of meta-knowledge has been explored, including (1) a generic feature embedding or metric space, in which images are easy to classify using a distance-based classifier such as cosine similarity or nearest neighbor (Koch et al., 2015; Vinyals et al., 2016; Snell et al., 2017; Sung et al., 2018; Ren et al., 2018; Oreshkin et al., 2018); (2) a common initialization of network parameters (Finn et al., 2017; Nichol and Schulman, 2018; Finn et al., 2018) or learned update rules (Andrychowicz et al., 2016; Ravi and Larochelle, 2017; Munkhdalai and Yu, 2017; Li et al., 2017; Rusu et al., 2019); (3) a transferable strategy to estimate model parameters based on few class examples (Bertinetto et al., 2016; Qiao et al., 2018; Qi et al., 2018; Gidaris and Komodakis, 2018), or from an initial small dataset model (Wang and Hebert, 2016; Wang et al., 2017). Some most recent work (Gidaris and Komodakis, 2018; Chen et al., 2019; Tian et al., 2020) also showed that the performance of these complex models can be matched by simple representation learning on base classes and classifier fine-tuning on novel classes.

# D DATASETS AND ADDITIONAL EXPERIMENT DETAILS

**Datasets: mini-ImageNet** consists of 64, 16, and 20 classes from ImageNet (Russakovsky et al., 2015) for base, validation, and novel sets, respectively. Each class contains 600 images of size $84 \times 84$. **tiered-ImageNet** consists of 351, 97, and 160 classes from ImageNet for base, validation, and novel sets, respectively. Unlike mini-ImageNet, the classes could have varying number of samples in tiered-ImageNet, but the images are of the same size. **CIFAR-FS** is a random split of CIFAR-100 (Krizhevsky and Hinton, 2009) with images of size $32 \times 32$ into 64, 16, and 20 classes for base, validation, and novel sets, respectively. **CUB** consists of 11,788 images of size $84 \times 84$ which are split into 100, 50, and 50 classes for base, validation, and novel sets, respectively.

**Setups and Implementation Details:** Our experiments fall into two main categories: (1) the standard ResNet (He et al., 2016) feature experiments for logistic classifiers, and (2) more powerful,

| Setting | Hyper-parameter | Value |
|---|---|---|
| All Standard Backbone Experiments | Learning Rate | 0.005 |
| | Mini-batch Size | 10 |
| | Number of Classes | 16 |
| | Optimizer | SGD |
| | Train-Heldout Splits | 90%-10% |
| | Backbones | ResNet 10, 18, 34, 50, 101 |
| Balanced Data Experiments in Sections 4.1 and 4.2 in the Main Paper | Number of Shots | 1, 5, 10, 15, 20, 25 |
| | Firth Coefficients Set* | 0, 0.01, 0.03, 0.1, 0.3, 1, 3, 10 |
| | L2 Coefficients Set** | 0, 1, 3, 10, 30, 100, 300, 1000 |
| Imbalanced Data Experiments in Section 4.3 in the Main Paper | 7.5-Shot Class Distribution | 2, 2, 2, 2, 4, 4, 4, 4, 8, 8, 8, 8, 16, 16, 16, 16 |
| | 15-Shot Class Distribution | 1, 1, 5, 5, 9, 9, 13, 13, 17, 17, 21, 21, 25, 25, 29, 29 |
| Experiments in Sections 4.1, 4.2, and 4.3 in the Main Paper with 1-Layer Logistic Classifier | Number of Epochs | 400 |
| | Classifier Architecture | Features→Classes→Softmax |
| | Number of Trials | More than 800 |
| Experiments in Section E.2 with 3-Layer Logistic Classifier | Number of Epochs | 100 |
| | Classifier Architecture | Features→100→ReLU→50 →ReLU →Classes→Softmax |
| | Number of Trials | More than 400 |

Table A3: Experimental settings used for the standard backbone experiments. The table is partitioned into 5 sections, where the first section shows the global hyper-parameters used in all standard backbone experiments. The same set of Firth bias reduction and L2 regularization coefficients were used for all validation experiments. *The Firth regularization coefficients were chosen for Equation (13) in the main paper. **We defined the L2-regularization as the **mean** squared value of all classifier parameters, which is why the normalized set of coefficients seems large. The typical unnormalized regularization coefficients can be obtained by dividing these normalized coefficients by the number of classifier parameters.

| Number of Shots | Confidence Penalty Improvements | Firth Improvements |
|---|---|---|
| 5 | $0.13 \pm 0.13$ % | $0.23 \pm 0.06$ % |
| 10 | $0.52 \pm 0.14$ % | $0.73 \pm 0.07$ % |
| 15 | $0.57 \pm 0.18$ % | $1.00 \pm 0.07$ % |

Table A4: **Comparing Firth bias reduction against the confidence penalty label smoothing technique.** The confidence penalty is defined as a $D_{\mathrm{KL}}(\mathrm{P}_i\|\mathrm{U})$ regularization term, whereas Firth bias reduction for logistic and cosine classifiers reduces to a $D_{\mathrm{KL}}(\mathrm{U}\|\mathrm{P}_i)$ penalty. The experimental setting is the same as Figure A7 with the ResNet-10 backbone. This suggests that the improvements obtained by Firth are the result of its bias suppression property, and Firth cannot be replaced by standard label smoothing techniques.

state-of-the-art WideResNet (Zagoruyko and Komodakis, 2016) features trained with strong regularization techniques (manifold mixup) and additional self-supervision (Mangla et al., 2020), or further calibrated via feature transformations (Yang et al., 2021), for logistic and cosine classifiers.

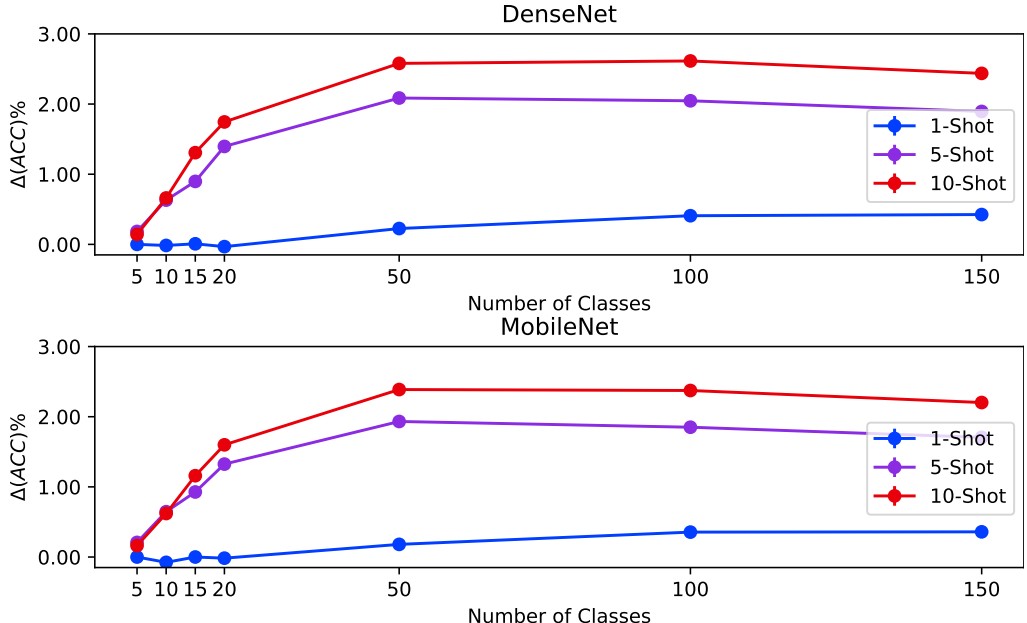

Figure A11: **Firth bias reduction improves the accuracy of the logistic classifier on tiered-ImageNet for both DenseNet (top) and MobileNet (bottom) as backbone architectures.** The improvement is consistent over the number of classes in few-shot classification. The error bars are almost non-existent (i.e., less than $0.02\%$), since over 10,000 trials with matching randomization factors were used for obtaining each single point. Firth bias reduction improvements are always positive, and it never hurts to use it.

For the first category of experiments, we averaged our results over 400 random trials. An array of 5 different ResNet architectures were trained in this category: ResNet10, 18, 34, 50, and 101, following a simple pipeline in the Pytorch library example[1]. It is worth noting that we deliberately did not engineer strong features for this category of experiments. The mini-ImageNet dataset was used for these experiments, and 16-way classification was performed on both the validation and novel classes. For this purpose, 16 out of 20 novel classes were chosen at random once and fixed for all the evaluations. We split the validation and novel classes into 90% training and 10% held-out for accuracy evaluation. The imbalanced data settings were chosen to either have 7.5 or 15 average number of shots per class. For the second category of experiments, we used the WideResNet-28-10 pre-trained feature backbones from Mangla et al. (2020); Yang et al. (2021). These experiments were conducted on three benchmarks, and each hyper-parameter configuration was averaged over 10,000 random trials. The backbone parameters and extracted features for all datasets are publicly available in the Illinois Data Bank repository (Saleh et al., 2022). Additional details and hyper-parameters are covered in the subsequent sections.

**Additional Implementation Details for Sections 4.1 and 4.2 in the Main Paper:** The L2-regularization coefficient was chosen for the mean-squared value of all classifier parameters, and the Firth bias reduction coefficient was chosen for Equation (13) in the main paper.

**Statistical Significance and Reducing the Effect of Randomized Factor:** Our study investigates improvements over the baseline. Random effects (random number seed; batch ordering; parameter initialization; and so on) complicate the study by creating variance in the measured improvement. We used a matching procedure (so that the baseline and the Firth penalized models share the same values of all random effects) to control this variance. As long as one does not search for random effects that yield a good improvement (we did not), this yields an unbiased estimate of the improvement. Each experiment is repeated multiple times to obtain confidence intervals. Note that (1) confidence intervals are small; and (2) improvements occur over a large range of feature backbones and datasets.

---

[1]https://github.com/pytorch/examples

| | | No Artificial Samples | | | 750 Artificial Samples | | |
|---|---|---|---|---|---|---|---|
| Way | Shot | Before | After | Improvement | Before | After | Improvement |
| 10 | 1 | 59.44 | $60.07 \pm 0.16$ | $0.63 \pm 0.04$ | 61.85 | $61.90 \pm 0.17$ | $0.05 \pm 0.02$ |
| 10 | 5 | 80.52 | $80.85 \pm 0.12$ | $0.33 \pm 0.03$ | 79.66 | $80.07 \pm 0.13$ | $0.42 \pm 0.04$ |
| 15 | 1 | 52.68 | $53.35 \pm 0.13$ | $0.67 \pm 0.03$ | 54.57 | $54.62 \pm 0.14$ | $0.05 \pm 0.02$ |
| 15 | 5 | 75.18 | $75.64 \pm 0.11$ | $0.46 \pm 0.03$ | 73.88 | $74.40 \pm 0.11$ | $0.53 \pm 0.04$ |

Table A5: **The Firth bias reduction accuracy improvements on the tiered-ImageNet dataset when 0 or 750 artificial samples were generated from the calibrated normal distribution in Yang et al. (2021).**

| | Before | After | Improvement | Before | After | Improvement |
|---|---|---|---|---|---|---|
| | | 5-way | | | 10-way | |
| 1-shot | 74.96 | $75.03 \pm 0.19$ | $0.07 \pm 0.01$ | 61.46 | $61.49 \pm 0.13$ | $0.03 \pm 0.00$ |
| 5-shot | 87.43 | $87.48 \pm 0.13$ | $0.06 \pm 0.00$ | 77.73 | $77.83 \pm 0.10$ | $0.10 \pm 0.00$ |
| 10-shot | 89.83 | $89.88 \pm 0.11$ | $0.05 \pm 0.00$ | 81.52 | $81.64 \pm 0.09$ | $0.11 \pm 0.00$ |
| | | 15-way | | | 20-way | |
| 1-shot | 53.45 | $53.47 \pm 0.10$ | $0.02 \pm 0.00$ | 47.78 | $47.79 \pm 0.07$ | $0.01 \pm 0.00$ |
| 5-shot | 70.70 | $70.99 \pm 0.07$ | $0.28 \pm 0.00$ | 65.26 | $65.60 \pm 0.03$ | $0.34 \pm 0.00$ |
| 10-shot | 75.37 | $75.71 \pm 0.06$ | $0.34 \pm 0.00$ | 70.57 | $70.99 \pm 0.02$ | $0.42 \pm 0.00$ |

Table A6: **The Firth bias reduction improvements on the CIFAR-FS dataset shown in Figure 4 in the main paper.** "Before" stands for the novel set accuracy without having any Firth bias reduction, and "After" stands for the novel set accuracy after applying Firth bias reduction. Note that the confidence intervals are much smaller for the improvement column, thanks to the random-effect matching procedure we used in this study. The "Before" confidence intervals were similar to the "After" confidence intervals, and thus not repeated due to space constraints.

It is safe to conclude that Firth bias reduction reliably offers a small but useful improvement in accuracy for few-shot classifiers.

**Additional Implementation Details for Section 4.3 in the Main Paper:** We used two non-uniform count vectors with different average counts, 7.5 (scheme 1) and 15 (scheme 2), to generate the datasets in both validation and novel sets. The count vector with the average of 7.5-shots had 4 classes for each count from the geometric sequence $\{2, 4, 8, 16\}$, and the count vector with the average of 15-shots had 2 classes for each count from the arithmetic sequence $\{1, 5, 9, 13, 17, 21, 25, 29\}$. The same 1-layer logistic classifier of Sections 4.1 and 4.2 was used in Section 4.3 in the main paper.

Table A3 summarizes the hyper-parameters used in all the standard backbone experiments. Also, Figure A8 shows the *relative* accuracy improvements corresponding to Figure A7. Figure A12 contains the validation accuracy versus Firth coefficient $\lambda$ for the experiments of Figure 2 and Figure A7.

**Additional Implementation Details for Section 4.5 in the Main Paper:** In the experiments carried out in Section 4.5 in the main paper, a 1-layer cosine classifier was used. Also, for the Firth bias-reduced cosine classifier, the regularization coefficient was tuned for each $(N, J)$ pair, with $N$ representing the number of samples per class and $J$ being the number of classes. For $J$-way classification on the novel set when $J$ happened to be larger than the number of classes in the validation set ($J_{\text{val}}$), the coefficient tuned for $J_{\text{val}}$-way classification was adopted.

| | Before | After | Improvement | Before | After | Improvement |
|---|---|---|---|---|---|---|
| | | 5-way | | | 10-way | |
| 1-shot | 65.17 | $65.59 \pm 0.18$ | $0.41 \pm 0.02$ | 50.38 | $50.64 \pm 0.11$ | $0.26 \pm 0.01$ |
| 5-shot | 82.60 | $83.04 \pm 0.12$ | $0.44 \pm 0.01$ | 71.15 | $71.91 \pm 0.10$ | $0.76 \pm 0.02$ |
| 10-shot | 86.82 | $87.04 \pm 0.09$ | $0.22 \pm 0.01$ | 77.34 | $77.87 \pm 0.08$ | $0.52 \pm 0.01$ |
| | | 15-way | | | 20-way | |
| 1-shot | 42.65 | $42.85 \pm 0.08$ | $0.20 \pm 0.01$ | 37.56 | $37.76 \pm 0.07$ | $0.20 \pm 0.00$ |
| 5-shot | 63.73 | $64.76 \pm 0.07$ | $1.03 \pm 0.01$ | 58.35 | $59.52 \pm 0.04$ | $1.17 \pm 0.01$ |
| 10-shot | 70.87 | $71.71 \pm 0.05$ | $0.84 \pm 0.01$ | 66.06 | $67.12 \pm 0.03$ | $1.06 \pm 0.01$ |

Table A7: **The Firth bias reduction improvements on the mini-ImageNet dataset shown in Figure 4 in the main paper.** "Before" stands for the novel set accuracy without having any Firth bias reduction, and "After" stands for the novel set accuracy after applying Firth bias reduction. Note that the confidence intervals are much smaller for the improvement column, thanks to the random-effect matching procedure we used in this study. The "Before" confidence intervals were similar to the "After" confidence intervals, and thus not repeated due to space constraints.

For the standard backbone experiments on mini-Imagenet, we trained more than 384,000 3-layer and 768,000 1-layer logistic classifiers for the balanced data settings. For the imbalanced settings on mini-Imagenet, we trained more than 64,000 3-layer and 128,000 1-layer logistic classifiers. For the cosine classifier experiments, we trained over 1.92, 1.92, and 3.36 million classifiers for mini-ImageNet, CIFAR-FS, and tiered-ImageNet datasets, respectively.

# E    ADDITIONAL EXPERIMENTS ON THE STANDARD BACKBONES

## E.1    ADDITIONAL LOGISTIC CLASSIFIER EXPERIMENTS

The experiments of Figure 2 were repeated to perform 16-way classification using a logistic classifier on tiered-ImageNet and CIFAR-FS in Figure A13. Moreover, 5-way and 10-way classification was tested for mini-ImageNet in Figure A6 in the same setting as Figure 2. The results show that Firth improvements always exist and it is even more effective as the number of classes increases.

## E.2    3-LAYER LOGISTIC CLASSIFIERS FOR THE STANDARD BACKBONES

We conducted the same experiments as in Sections 4.1 and 4.2 in the main paper but with a 3-layer logistic classifier. Again, we see consistent accuracy improvements for the Firth bias-reduced classifier over the non-penalized (baseline) classifier in Figure A7. This further supports the idea that *Firth bias reduction boosts the performance of any reasonable classifier*. Needless to say, L2-regularization is not effective as shown in Figure A7.

Furthermore, the imbalanced few-shot classification in Section 4.3 in the main paper was repeated with the 3-layer logistic classifier in Figure A10. Again for both schemes, the Firth penalized classifier has larger accuracy improvement than the classifier penalized with the KL-divergence to the non-uniform prior over the class probabilities. This further validates the effectiveness of Firth bias reduction in reducing the parameter estimation bias present in the few-shot setting.

# F    ADDITIONAL FEATURE BACKBONES

To test the Firth bias reduction technique for additional backbones, we used pre-trained DenseNet and MobileNet backbones on tiered-ImageNet from Wang et al. (2019). The accuracy improvements of Firth penalized logistic classifier over the baseline averaged over 10,000 trials are plotted in Figure A11. Regardless of the number of classes, the improvements are always positive.

| | Before | After | Improvement | Before | After | Improvement |
|---|---|---|---|---|---|---|
| | | 5-way | | | 10-way | |
| 1-shot | 73.50 | $73.64 \pm 0.25$ | $0.14 \pm 0.03$ | 61.20 | $61.44 \pm 0.16$ | $0.24 \pm 0.01$ |
| 5-shot | 88.00 | $88.31 \pm 0.12$ | $0.30 \pm 0.01$ | 79.41 | $80.01 \pm 0.11$ | $0.60 \pm 0.01$ |
| 10-shot | 90.94 | $91.14 \pm 0.10$ | $0.21 \pm 0.01$ | 83.88 | $84.47 \pm 0.09$ | $0.58 \pm 0.01$ |
| | | 15-way | | | 20-way | |
| 1-shot | 53.90 | $53.97 \pm 0.15$ | $0.07 \pm 0.01$ | 48.81 | $48.96 \pm 0.11$ | $0.15 \pm 0.01$ |
| 5-shot | 73.33 | $74.21 \pm 0.09$ | $0.88 \pm 0.01$ | 68.58 | $69.71 \pm 0.08$ | $1.13 \pm 0.01$ |
| 10-shot | 78.70 | $79.57 \pm 0.08$ | $0.86 \pm 0.01$ | 74.58 | $75.74 \pm 0.07$ | $1.16 \pm 0.01$ |
| | | 50-way | | | 100-way | |
| 1-shot | 33.91 | $34.13 \pm 0.06$ | $0.22 \pm 0.01$ | 24.80 | $25.00 \pm 0.03$ | $0.20 \pm 0.00$ |
| 5-shot | 52.60 | $54.71 \pm 0.05$ | $2.10 \pm 0.01$ | 41.03 | $43.59 \pm 0.03$ | $2.56 \pm 0.01$ |
| 10-shot | 59.67 | $61.94 \pm 0.04$ | $2.27 \pm 0.01$ | 47.77 | $50.81 \pm 0.02$ | $3.04 \pm 0.01$ |
| | | 150-way | | | | |
| 1-shot | 20.37 | $20.56 \pm 0.02$ | $0.19 \pm 0.00$ | | | |
| 5-shot | 34.89 | $37.54 \pm 0.01$ | $2.65 \pm 0.01$ | | | |
| 10-shot | 41.04 | $44.28 \pm 0.01$ | $3.25 \pm 0.01$ | | | |

Table A8: **The Firth bias reduction improvements on the tiered-ImageNet dataset shown in Figure 4 in the main paper.** "Before" stands for the novel set accuracy without having any Firth bias reduction, and "After" stands for the novel set accuracy after applying Firth bias reduction. Note that the confidence intervals are much smaller for the improvement column, thanks to the random-effect matching procedure we used in this study. The "Before" confidence intervals were similar to the "After" confidence intervals, and thus not repeated due to space constraints.

# G    COMPARING FIRTH BIAS REDUCTION AGAINST STANDARD LABEL SMOOTHING TECHNIQUES

To demonstrate that Firth bias reduction cannot simply be replaced with label smoothing, we tested two advanced variants of label smoothing that are superior to the original version as proposed by Pereyra et al. (2017). The first variant, called confidence penalty, uses the entropy of the classifier's output (or equivalently, reverses the direction of the KL divergence in the original version of label smoothing (Szegedy et al., 2016)); and the second variant, called unigram label smoothing, uses prior distribution over the classes instead of uniform, which has been shown to be advantageous when the output labels' distribution is imbalanced in Pereyra et al. (2017). Note that both variants were investigated for training a full deep neural network with a feature extractor backbone in large-sample regimes in Pereyra et al. (2017). Our experiments in Figure 3 evaluate the effect of unigram label smoothing when training the classifier in the small-sample regime.

We also performed more experiments in the same setting to compare Firth bias reduction against confidence penalty regularization. As summarized in Table A4, in all the settings Firth bias reduction has larger significant improvements than the confidence penalty technique. This further supports the value of using Firth bias reduction and the fact that its impact on few-shot classification cannot be reproduced with well-known and widely-used label smoothing techniques.

| | Before | After | Improvement | Before | After | Improvement |
|---|---|---|---|---|---|---|
| | | ResNet10 | | | ResNet18 | |
| 1-shot | 7.96 | $7.97 \pm 0.06$ | $0.01 \pm 0.01$ | 7.57 | $7.58 \pm 0.05$ | $0.01 \pm 0.01$ |
| 5-shot | 8.22 | $8.23 \pm 0.06$ | $0.01 \pm 0.01$ | 7.94 | $7.95 \pm 0.06$ | $0.01 \pm 0.01$ |
| 10-shot | 8.19 | $8.24 \pm 0.05$ | $0.06 \pm 0.05$ | 8.01 | $8.15 \pm 0.05$ | $0.14 \pm 0.06$ |
| 15-shot | 8.21 | $8.36 \pm 0.05$ | $0.15 \pm 0.06$ | 8.09 | $8.47 \pm 0.05$ | $0.38 \pm 0.06$ |
| 20-shot | 8.22 | $8.51 \pm 0.05$ | $0.28 \pm 0.06$ | 8.15 | $8.75 \pm 0.06$ | $0.60 \pm 0.06$ |
| 25-shot | 8.25 | $8.51 \pm 0.05$ | $0.25 \pm 0.06$ | 8.27 | $8.92 \pm 0.05$ | $0.65 \pm 0.06$ |
| | | ResNet34 | | | ResNet50 | |
| 1-shot | 7.47 | $7.48 \pm 0.05$ | $0.01 \pm 0.01$ | 7.51 | $7.52 \pm 0.05$ | $0.01 \pm 0.01$ |
| 5-shot | 7.69 | $7.70 \pm 0.05$ | $0.01 \pm 0.01$ | 7.73 | $7.78 \pm 0.05$ | $0.05 \pm 0.05$ |
| 10-shot | 7.73 | $7.96 \pm 0.05$ | $0.23 \pm 0.05$ | 7.83 | $8.59 \pm 0.05$ | $0.76 \pm 0.06$ |
| 15-shot | 7.79 | $8.22 \pm 0.05$ | $0.43 \pm 0.06$ | 7.89 | $9.02 \pm 0.05$ | $1.13 \pm 0.06$ |
| 20-shot | 7.87 | $8.41 \pm 0.05$ | $0.55 \pm 0.06$ | 8.15 | $9.67 \pm 0.06$ | $1.52 \pm 0.06$ |
| 25-shot | 7.97 | $8.54 \pm 0.05$ | $0.57 \pm 0.06$ | 8.44 | $10.71 \pm 0.06$ | $2.27 \pm 0.06$ |
| | | ResNet101 | | | | |
| 1-shot | 7.63 | $7.65 \pm 0.05$ | $0.02 \pm 0.02$ | | | |
| 5-shot | 7.81 | $7.88 \pm 0.05$ | $0.07 \pm 0.05$ | | | |
| 10-shot | 7.96 | $8.81 \pm 0.05$ | $0.86 \pm 0.06$ | | | |
| 15-shot | 8.10 | $9.50 \pm 0.05$ | $1.40 \pm 0.06$ | | | |
| 20-shot | 8.37 | $10.29 \pm 0.06$ | $1.91 \pm 0.06$ | | | |
| 25-shot | 8.74 | $10.62 \pm 0.06$ | $1.88 \pm 0.06$ | | | |

Table A9: **The Firth bias reduction improvements on the mini-ImageNet dataset shown in Figure 2 in the main paper.** "Before" stands for the novel set accuracy without having any Firth bias reduction, and "After" stands for the novel set accuracy after applying Firth bias reduction. Note that the confidence intervals are much smaller for the improvement column, thanks to the random-effect matching procedure we used in this study. The "Before" confidence intervals were similar to the "After" confidence intervals, and thus not repeated due to space constraints. It is worth noting that we deliberately did not engineer strong features for this experiment (stronger feature backbone results are shown in Sections 4.5 and 4.6 in the main paper). This diversifies Firth's performance portfolio, demonstrating its robustness to the strength of the feature backbones; even with weak features, Firth bias reduction substantially improves the accuracy with high relative improvements as shown here and in Figure A8.

# H    ADDITIONAL COMPARISON WITH STATE OF THE ART

Table A5 summarizes the accuracy improvements obtained by integrating Firth bias reduction with the distribution calibration method (Yang et al., 2021) under different shots and ways on the tiered-ImageNet dataset. This method calibrates the features to follow a normal distribution, and generates artificial samples from the estimated normal distribution as data augmentation to aid few-shot classification. In its state-of-the-art setting, the features are transformed using Tukey transformations, 750 artificial samples are generated per class, and a logistic classifier is used. We tested Firth bias reduction in two scenarios: (1) state-of-the-art setting without generating artificial samples from the calibrated distribution; and (2) state-of-the-art setting with 750 artificial samples generated per class,

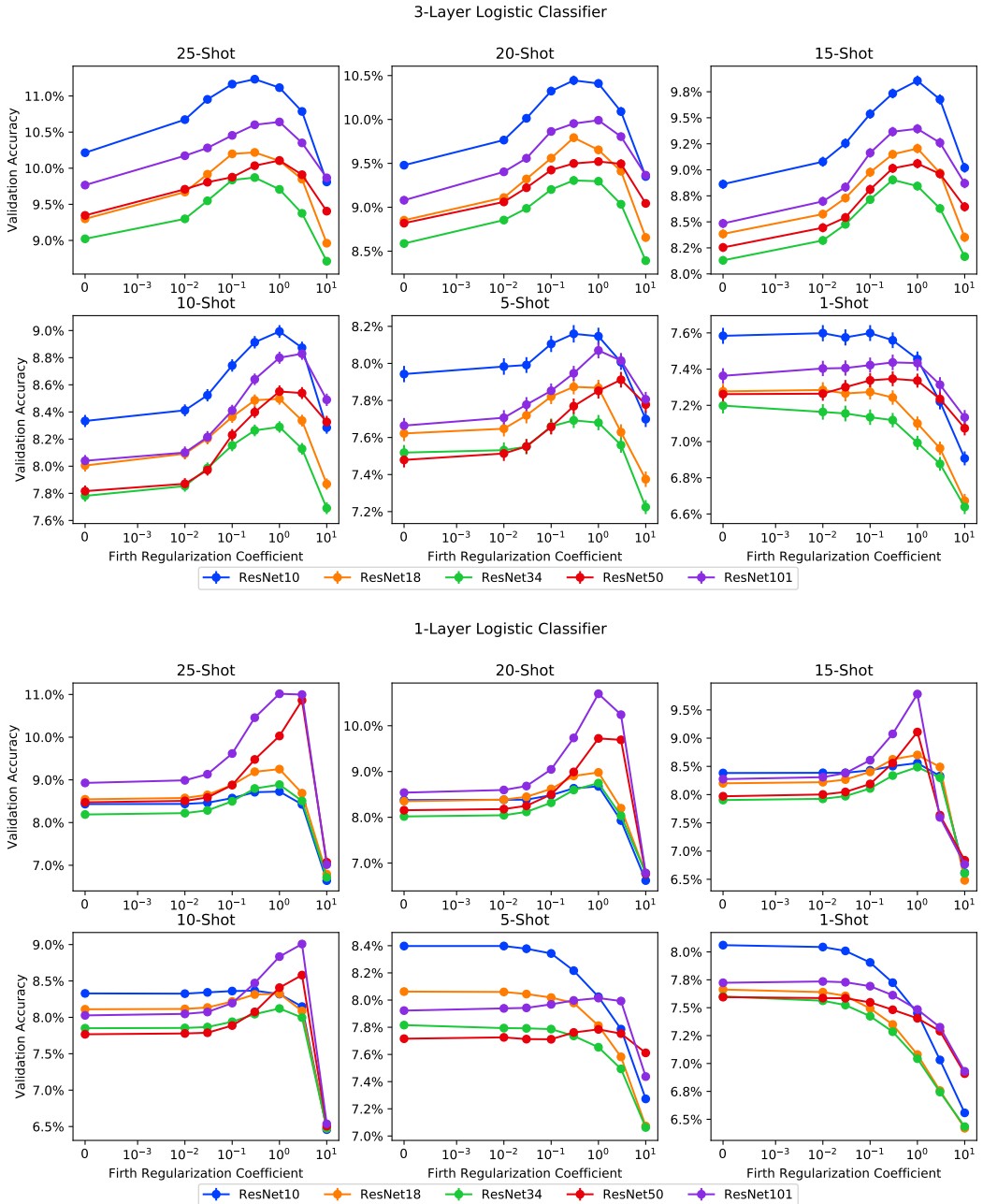

Figure A12: **The effect of the validation coefficient $\lambda$ on the validation accuracy for different number of shots and backbone architectures.** The top two rows belong to the 3-layer logistic classifier in Figure A7 in the Appendix, and the bottom two rows belong to the 1-layer logistic classifier in Figure 2 in the main paper.

as shown in Table A5. The $95\%$ confidence intervals for the accuracy improvements are reported in both cases.

As shown in Table A5, Firth bias reduction produces positive improvements in all cases, which are statistically significant in all the cases. As expected, Firth bias reduction produces larger improvements when artificial sample generation is disabled, and thus there is a larger maximum likelihood estimation bias on the logistic classifier (despite the use of Tukey transformation). In the case that artificial samples are added and they are tuned to follow the original normal distribution, having 750

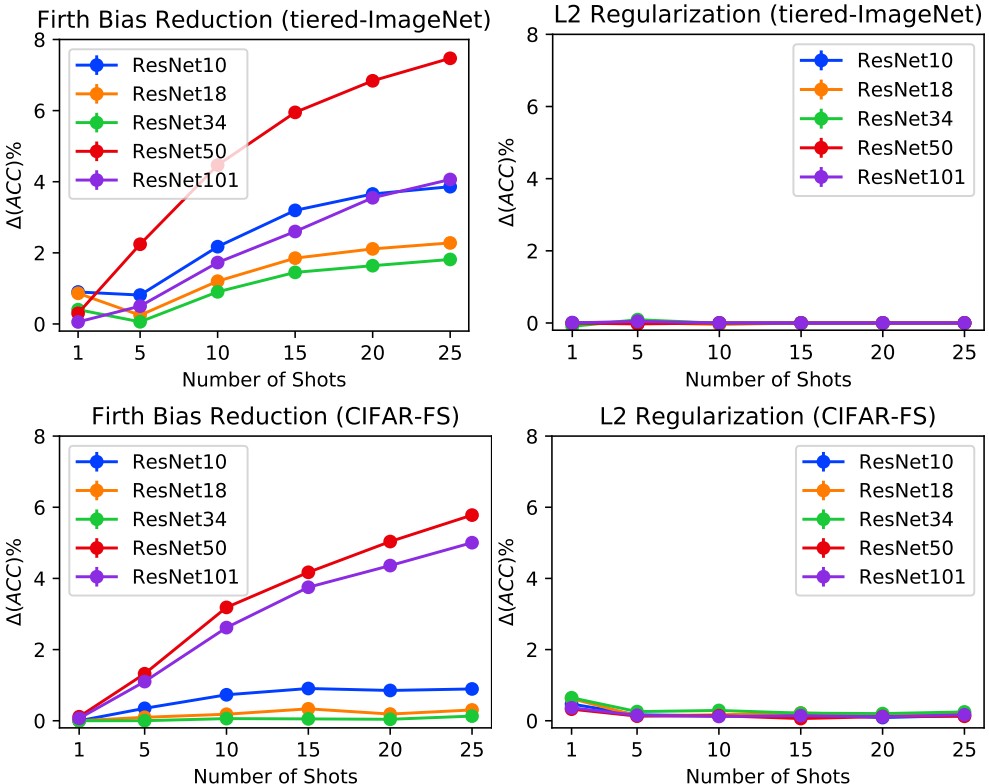

Figure A13: **Comparing the effect of Firth bias reduction against L2-regularization for the novel set classification accuracy on the tiered-ImageNet and CIFAR-FS datasets**. Firth bias reduction (**left**) delivers a novel class accuracy improvement over the baseline classifier up to 8.0%. By contrast, L2-regularization (**right**) is mostly statistically insignificant in the same few-shot setting. **16-way** logistic classification was conducted in this experiment, with over 1,000 randomized and matching trials for each combination of method, backbone, and number of samples.

of them can to some extent alleviate the bias in the estimation of the logistic classifier's weights in the few-shot regime. However, the results show that even in the presence of more data (artificial samples), Firth bias reduction is still effective. This suggests that producing artificial samples to augment data cannot resolve the estimation bias issue, as they are more likely to be similar to the limited real samples available. As shown in Section 4.6, this becomes an even more severe problem in the cross-domain few-shot setting; producing artificial samples is significantly less effective than Firth bias reduction, due to the domain shift.

