# OpenReview forum: "On the Importance of Firth Bias Reduction in Few-Shot Classification"
_ICLR.cc/2022/Conference — ICLR 2022 Spotlight_

### Official Review · Reviewer_s811 · 2021-10-20

**Correctness:** 4
**Technical Novelty And Significance:** 3
**Empirical Novelty And Significance:** 3
**Recommendation:** 6
**Confidence:** 4

**Main Review:**

This paper has multiple advantages.
-This is the first paper addressing the effectiveness of Firth bias reduction in few-shot learning.
-Vast empirical study validates the proposed method consistently improves the performance of the baseline method, and no performance penalty is introduced.
-Apart from typical logistic regression models, the firth bias reduction can also be applied to cosine classifiers.
-Comprehensive discussion about other regularization techniques is provided.

I also have several questions and concerns about this paper.
-The relative improvements are validated through empirical investigations. I expect to see the theoretical guarantee for the improvement of firth bias reduction in few-shot learning.
-The idea of this paper partially comes from [1], which weakens the novelty of this manuscript. I would suggest the authors discuss more the extensions of [1] in current machine learning tasks, and discuss the difference with them.
-All the experimental metrics are reported by relative improvement, making it hard to be compared for other upcoming works. I suggest the authors report the absolute accuracy on benchmark datasets in the supplementary material.

[1] D. Firth. Bias reduction of maximum likelihood estimates. Biometrika, 80(1):27–38, 1993


**Summary Of The Paper:**

This paper tackles the few-shot learning problem, which is a hot topic in the representation learning field. Specifically, the authors aim to train accurate classifiers using a small number of samples – Maximum Likelihood Estimators are biased for them. As a result, the authors propose to utilize Firth’s Penalized Maximum Likelihood Estimator to modify the ordinary MLEs, and prevent the bias when training classifiers for few-shot instances. The authors have validated the effectiveness of Firth bias reduction from multiple aspects, and the experimental results show the performance improvement is consistent across various circumstances, e.g., with different backbones, data distributions, and classifiers.

**Summary Of The Review:**

To conclude, this paper is clearly elaborated with sufficient experimental results and promising improvement. I suggest a ‘weak accept.’

---

> ### Author Response · Authors · 2021-11-23
> **Response to Reviewer s811**
>
> **Question 1:**
> The relative improvements are validated through empirical investigations. I expect to see the theoretical guarantee for the improvement of firth bias reduction in few-shot learning. -The idea of this paper partially comes from [1], which weakens the novelty of this manuscript. I would suggest the authors discuss more the extensions of [1] in current machine learning tasks, and discuss the difference with them.
>
> **Response:**
>
> In the original work by [1], the Firth bias reduction term is introduced and proved to be effective for the broad category of distributions that belong to the exponential family. Generalized linear models that are popular and frequently used in machine learning tasks are in this family.
>
> Furthermore, Firth has proven to be useful in applications where separation happens which is the case for small sample sizes. In classification, separation means when a subset of covariates can correctly categorize the sample points. It is shown that separation is necessary and sufficient to make maximum likelihood estimates numerically unstable and infinite [2]. On the other hand, Firth’s PMLE provides finite bias-reduced estimates in the presence of separation [3,4] which makes it to be widely used in applied work.
>
> To reiterate some of the theoretical contributions of our work (which is not in [1]), we extended the Firth bias reduction technique to the few-shot settings where the number of samples is so small that the estimated FIM (Fisher Information Matrix) is low-rank, making the $\log(\det(F))$ undefined. To mitigate this, we amended the definition of the determinant to have a well-defined Firth bias reduction term under few-shot scenarios. We applied a series of steps to simplify the $\log(\det(F))$ form as a KL to uniform formulation in the special case of logistic regression models in Appendix A; this provided a way for efficiently computing the Firth bias reduction term. Furthermore, we theoretically proved that the KL to uniform formulation of Firth bias reduction extends to the cosine classifiers as well in Appendix B.
>
> Unfortunately, we did not find further relevant work on applying Firth bias reduction to other machine learning tasks (after a relatively comprehensive literature review), but we would be very happy to discuss if there is any.
>
> **Question 2:**
> All the experimental metrics are reported by relative improvement, making it hard to be compared for other upcoming works. I suggest the authors report the absolute accuracy on benchmark datasets in the supplementary material.
>
> **Response:**
> We thank the reviewer for the suggestion. Appendix Tables A6, A7, A8, and A9 containing absolute and delta accuracies before and after applying Firth bias reduction are provided for all the settings in Figure 2 and Figure 4.

---

### Official Review · Reviewer_4kx4 · 2021-10-28

**Correctness:** 3
**Technical Novelty And Significance:** 4
**Empirical Novelty And Significance:** 4
**Recommendation:** 8
**Confidence:** 5

**Main Review:**

# Main review

This paper is clearly written with extensive experiments to support the proposed approach.

The strength of this paper is related to the effective computation of the Firth bias reduction term resulting to a simple pragmatical implementation. This pragmatical implementation turns out to be a theoretical justification of the Label smoothing proposed in Szegedy et al.

In the mathematical notations the logistic regression weights are defined so that $\beta_0=0$. This allows to have a non-redundant notation for logistic regression weights and to easily carry out computation of the loss associated to Firth bias reduction. However it came with some drawbacks:
- when using L2 regularization, the regularization could be non optimal. What about the impact here on the performance of L2 regularization? (from table A2, we can also observe that the set of weights for L2 regularization is quite high - usually when using L2 regularization we observe the use of weights of order 10-4 ... 10-2)
- when extending the computation of the Firth Loss to cosine classifiers, logistic regression weights are normalized... this is not compatible with the setting $\beta_0=0$. The associated demonstration needs to be revisited. (although intuitively if we omit this constraint on notations, we should be able to redo the same reasoning)

Extensive experiments clearly show the benefits of using the proposed bias reduction. To complement those results, it would be also interesting to have a final table with resuming the newly obtained SOTA results.

Also it would be interesting to have an idea on the effective value of $\lambda$ to be used. Table A.2 shows a large set of values there. Could the author share what are the typical values observed on some configuration tested? (e.g. for figure 3?). Also what is the sensitivity of $\lambda$ on the performance?

# Minor remarks
- equation 1: $j$ is used in the summation in the denominator... it should be changed to e.g. $j'$
- just after equation (10), it should be mentioned in the text what is $\lambda$ used for.
- annex B: in the definition of the C term, $\lambda$ should appear


**Summary Of The Paper:**

The paper proposes to reduce the bias inherent to the estimation of the classifier used in Few Shot Classification settings. To this purpose Firth bias reduction is proposed to be used. Theoretical computation of this bias reduction is carried out within the context of Few Shot Classification resulting to a practical implementation. It turns out that it could be associated to a regularization term encouraging uniform class assignment probabilities. This results is also generalized to the context of using a cosine classifier on the features.

Extensive experiments are carried out to show the benefits of this bias reduction on SOTA approaches and with various backbone architectures.

**Summary Of The Review:**

Overall I am in favour of accepting this paper. The strength of this paper is on the theoretical justification of the proposed regularization term introduced associated to the general Firth bias reduction. Also extensive results are carried out to evaluate fairly the benefits of the proposed approach on SOTA solution.

---

> ### Author Response · Authors · 2021-11-23
> **Response to Reviewer 4kx4 - Part 2**
>
> **Question 2:**
> Extensive experiments clearly show the benefits of using the proposed bias reduction. To complement those results, it would be also interesting to have a final table with resuming the newly obtained SOTA results.
>
> **Response:**
> Per reviewer's request, we have added more experiments with the recent state-of-the-art distribution calibration (DCL) method [7]. We show that the Firth bias reduction term can be incorporated into DCL and consistently improve its performance in different settings. In Appendix Section H and Table A5, we summarize the improvements on tiered-ImageNet. More importantly, in the main submission, we show the results in the **cross-domain setting** which is more challenging, and record the improvements achieved by Firth in Table 1. For the cross-domain setting between dataset 1 and dataset 2, the pretrained feature backbone on dataset 1 was used to get the features on the novel set of dataset 2. mini-ImageNet or tiered-ImageNet was used as dataset 1, and CUB was used as dataset 2 in our cross-domain experiments. In all the settings, Firth provides significant improvements.
>
> **Question 3:**
> Also it would be interesting to have an idea on the effective value of λ to be used. Table A.2 shows a large set of values there. Could the author share what are the typical values observed on some configuration tested? (e.g. for figure 3?). Also what is the sensitivity of λ on the performance?
>
> **Response:**
> We have added the validation accuracy versus Firth coefficient $\lambda$ plots in Figure A12 for all the settings of experiments in Figure 2 and Figure A5 on mini-ImageNet. Based on these plots, the peak performance mostly happens for $\lambda \in \\{0.3, 1, 3\\}$, however this range might vary based on the application.
>
> **Question 4:**
> Minor remarks
>
> equation 1: j is used in the summation in the denominator... it should be changed to e.g. j′
>
> just after equation (10), it should be mentioned in the text what is λ used for.
>
> Appendix B: in the definition of the C term, λ should appear
>
> **Response:**
> We thank the reviewer for spotting these errors. All of the corrections have been made as suggested by the reviewer.

---

> > ### Comment · Reviewer_4kx4 · 2021-11-26
> > **updated review**
> >
> > Thanks to the authors for these clarifications and complements. There are no more issues for me, so I keep my score unchanged as accept.

---

> ### Author Response · Authors · 2021-11-23
> **Response to Reviewer 4kx4 - Part 1**
>
> **Question 1:**
> In the mathematical notations the logistic regression weights are defined so that β0=0. This allows to have a non-redundant notation for logistic regression weights and to easily carry out computation of the loss associated to Firth bias reduction. However it came with some drawbacks:
>
> when using L2 regularization, the regularization could be non optimal. What about the impact here on the performance of L2 regularization? (from table A2, we can also observe that the set of weights for L2 regularization is quite high - usually when using L2 regularization we observe the use of weights of order 10-4 ... 10-2)
>
> **Response:**
> We appreciate the reviewer's valuable comment and minute attention to the details (including the experimental details).
>
> The $\beta_0=0$ is mainly a common theoretical assumption made without loss of generality, and it is only done in the theoretical setup since it simplifies the derivation of derivatives. **To be clear, we never set $\beta_0=0$ in any of our experiments, and the same set of parameters were used for both Firth bias reduction and L2-regularization**. This was done since the final form of the Firth bias reduction term $\text{CE}(U_{[0,J]}||P_i)$ does not require imposing such a constraint on the classifier, and can be computed using any set of parameters. In theory, any set of non-constrained logistic weights $(\beta_0, \beta_1, \cdots, \beta_J)$ can be converted to a constrained set of logistic weights $(0, \beta_1-\beta_0, \beta_2-\beta_1, \cdots \beta_J-\beta_0)$ which produces identical output to the non-constrained set (i.e., any baseline shift in the logistic weights will be cancelled out in the output of the softmax function). This is why the Firth theoretical derivations can easily extend to any set of non-constrained logistic parameters $(\beta_0, \beta_1, \cdots, \beta_J)$ without any modification to the Firth bias reduction term.
>
> As for the range of L2 regularization coefficients, we should note that **we normalized the L2-regularization term by the total number of classifier parameters, which explains why our regularization coefficients are seemingly large whereas they are not**. In other words, instead of defining the L2-regularization term as $\lambda_{\text{L2}} \sum_{k=1}^{d}w_k^2$, we defined it as $\frac{\lambda_{\text{L2}}}{d}\sum_{k=1}^{d}w_k^2$. This was only done since **our array of experiments included a large range of number of parameters in the classifier**. For instance, a ResNet101 backbone in a 25-way classification task has roughly **20 times more classifier parameters** than a ResNet10 on a 5-way classification task in Figure 2. Such a stark difference in the number of parameters makes the proper range of L2 regularization coefficients incompatible for different settings. However, by normalizing the L2-regularization term by the number of parameters, all settings could share the same set of coefficient candidates.
>
> **To be clear, our effective L2-regularization coefficient orders are consistent with the reviewer's suggestion**. For instance, a typical ResNet50 on a 15-way classification task has roughly 30,000 classifier parameters. For this particular case plugging $\lambda_{\text{L2}} \in [1, 1000]$ into $\frac{\lambda_{\text{L2}}}{d} \sum_{k=1}^{d}w_k^2$ is equivalent to plugging $\lambda_{\text{L2}} \in [3.3\times 10^{-5}, 3.3\times 10^{-2}]$ into $\lambda_{\text{L2}} \sum_{k=1}^{d}w_k^2$.
>
> To clarify this note, we added the following sentence to the caption of the hyper-parameter table:
>
> *"We defined the L2-regularization as the **mean** squared value of all classifier parameters, which is why the normalized set of coefficients seems large. The typical unnormalized regularization coefficients can be obtained by dividing these normalized coefficients by the number of classifier parameters."*

---

### Official Review · Reviewer_37fi · 2021-11-01

**Correctness:** 4
**Technical Novelty And Significance:** 4
**Empirical Novelty And Significance:** 3
**Recommendation:** 8
**Confidence:** 3

**Main Review:**

Q1- According to the intro, which I quote here ``We achieve this by deriving a simplified yet effective Firth formulation that penalizes the KL-divergence between the predicted distribution and the uniform distribution of classes'', the proposed solution assumes that the distribution of query samples is uniform. One wonders how restrictive this assumption might be, can authors comment on this? I understand that in FSL, current protocols have that property but when it comes to learning, many algorithms do not make such an assumption.
I have to add that I understand that authors performed experiments in section 4.3 about imbalanced datasets and discussed replacing the uniform prior with non-uniform one, but from a theoretical point of view, is it possible to provide more insights?

Q2- Based on Fig.1, it seems that the proposed method works better if number of shots is large, and specifically for very deep models. Would be very useful if authors can provide some insights about this.

Q3- I can see some sort of inconsistency in the experiments. It would be good to see why authors didn't use the same datasets in Fig.3 for experiments in Fig.1. Also, the choice of 16class in Fig.1 is not well-justified. How does the method work if less classes were considered?

Q4- Out of curiosity, did authors evaluate their method in conjunction with other FSL techniques, say other embedding techniques?

A suggestion,  it is very hard for me to fully read the improvements in Fig.3, why not simply providing a table so the reader can better understand the improvements?



**Summary Of The Paper:**

This work introduces a method to reduce the bias of MLE for FSL problems. Authors developed the bias reduction formulation based on the Firth bias reduction method and extended it for cosine classifier as well. A through set of experiments demonstrate that the proposed method provides statistically significant improvement over the baselines.

**Summary Of The Review:**

I believe the proposed method is novel and interesting.

---

> ### Author Response · Authors · 2021-11-23
> **Response to Reviewer 37fi - Part 2**
>
> **Question 3:**
> I can see some sort of inconsistency in the experiments. It would be good to see why authors didn't use the same datasets in Fig.3 for experiments in Fig.1. Also, the choice of 16class in Fig.1 is not well-justified. How does the method work if less classes were considered?
>
> **Response:**
> Figure 4 investigates broader settings than Figure 2 -- more datasets, various ways, and various shots. To explore the impact of Firth on more datasets in the setting of Figure 2, we repeated the experiments of Figure 2 for tiered-ImageNet and CIFAR-FS. The plots are included in Appendix Figure A13. Firth consistently improves the performance for these two datasets on the standard ResNet backbones as well.
>
> Initially we intended to test Firth on the hardest classification task (with all the classes) that was possible on mini-ImageNet. The highest number of classes was restricted by the number of validation classes which is 16 on mini-ImageNet. Note that Firth can be applied to different classification tasks with varied number of novel classes (i.e., number of ways). Per reviewer's request, we repeated the experiments of Figure 2 on mini-ImageNet for 5-way and 10-way classification tasks and included the results in Appendix Figure A9. These results suggest that Firth bias reduction is consistently effective across all the settings, and that its effectiveness is more pronounced in harder classification tasks, which is in line with the observation of Figure 4.
>
> **Question 4:**
> Out of curiosity, did authors evaluate their method in conjunction with other FSL techniques, say other embedding techniques?
>
> **Response:**
> In the original submission, we have investigated a variety of feature backbones that are widely-used in the current few-shot learning tasks (various ResNets with varying depth, WideResNet trained with strong manifold mixup regularization, the embedding from the state-of-the-art Distribution Calibration technique [7]).
>
> Furthermore, per reviewer's request, we have added the improvements of Firth penalized logistic classifier using two additional feature backbones -- the SimpleShot embedding [6] with DenseNet or MobileNet -- on tiered-ImageNet in Appendix Figure A10. Similar to Figure 4, we tested the improvements of Firth penalized logistic classifier for a wide range of classification tasks. In line with the evidences of the original submission, Firth improves the classification performance regardless of the feature backbone/embedding, number of shots, and number of classes.
>
>
> **Question 5:**
> A suggestion, it is very hard for me to fully read the improvements in Fig.3, why not simply providing a table so the reader can better understand the improvements?
>
> **Response:**
> We plotted curves to easily show how performance changes w.r.t. different settings. We thank the reviewer for the suggestion, and we realized that improvements are hard to read on the plots, especially for the 1-shot setting. Therefore we have included Appendix Tables A6, A7, and A8, containing the absolute and delta accuracy values for Figure 4.

---

> ### Author Response · Authors · 2021-11-23
> **Response to Reviewer 37fi - Part 1**
>
> **Question 1:**
> According to the intro, which I quote here "We achieve this by deriving a simplified yet effective Firth formulation that penalizes the KL-divergence between the predicted distribution and the uniform distribution of classes", the proposed solution assumes that the distribution of query samples is uniform. One wonders how restrictive this assumption might be, can authors comment on this? I understand that in FSL, current protocols have that property but when it comes to learning, many algorithms do not make such an assumption. I have to add that I understand that authors performed experiments in section 4.3 about imbalanced datasets and discussed replacing the uniform prior with non-uniform one, but from a theoretical point of view, is it possible to provide more insights?
>
> **Response:**
> First, we would like to emphasize that the main contribution of the Firth technique is its bias reduction property for MLEs. Firth’s penalty term simplifies to encouraging uniform distribution over classes in the special case of logistic regression models – notice that we did not impose any assumption that the distribution of query samples is uniform; and empirically our method works on imbalanced datasets as well. In the following we provide insight from a different perspective on why such simplification should not be considered as restrictive but useful to achieve better performance in typical few-shot tasks.
>
> 1.	According to Section 3.2 of [5], MLE leads to severe overfitting in training complex models with limited data. The overfitting restricts the model’s flexibility to capture the true trends of the larger population, to which the limited training samples belong. That is, the MLE mimics the label distribution of the training samples closely, which may not be ideal. Therefore, when one chooses to use MLE on few training samples, there is always the risk of overfitting to the training set distribution and not performing well on the larger population that it is sampled from.
> 2.	In general, regardless of the true distribution of the larger population being uniform or imbalanced, a small set of samples can barely carry reliable information on the true distribution. The distribution of a small sample set can be highly variable and may even have a significant bias away from the true distribution.
> 3.	Given 1 and 2, when working with MLE using a small sample set, it is best to help MLE to be less misguided while training, by encouraging uniform distribution over the classes.
>
>
> **Question 2:**
> Based on Fig.1, it seems that the proposed method works better if number of shots is large, and specifically for very deep models. Would be very useful if authors can provide some insights about this.
>
> **Response:**
> Firth bias reduction only removes the leading $O(N^{-1})$ term from the MLE bias. In general, with fewer samples, the bias becomes a bigger issue, and the removal of $O(N^{-1})$ term may not be enough to guarantee better parameter estimations and performance. How to better cope with this extremely-few-shot regime has been a general challenge in few-shot learning, and is an interesting future direction.
>
> In addition, the observation of larger Firth improvements for deeper backbones in Figure 2 on mini-ImageNet does not hold for the same experiments on tiered-ImageNet in Figure A13, where the ResNet 101 improvement is closer to ResNet10 than ResNet50. However, based on all of the experiments we have done so far, while Firth is consistently effective, it seems that the specific behavior of Firth bias reduction may depend on a variety of factors, including feature distribution and classifier's complexity.

---

### Official Review · Reviewer_gyef · 2021-11-01

**Correctness:** 4
**Technical Novelty And Significance:** 3
**Empirical Novelty And Significance:** 3
**Recommendation:** 8
**Confidence:** 3

**Main Review:**

I enjoyed reading the submission. It's clear, well-written, and it explores a simple idea with rigor. The proposed idea is theoretically grounded and the experimental results are convincing.

Strengths:
- Clarity, writing quality.
- Novelty: as far as I know, the issue of MLE's bias in the finite-data regime has not been discussed in the few-shot classification literature.
- Rigor: the proposed approach is compared against sensible baselines (L2 regularization, label smoothing), controls for randomized factors, computes and compares confidence intervals, and considers multiple benchmarks and backbone architectures.

Weaknesses:
- The submission goes a bit quickly over the small-sample bias of MLE and Firth bias reduction for MLE. Without going into complete proofs, sketching the main arguments in both cases would make the submission accessible to a wider audience. As it stands, I feel like the submission is fully clear only to readers with prior knowledge of these notions.

Additional questions/comments:
- Can the authors elaborate on why the number of novel classes subsampled for Figure 1 is 16, specifically?
- In Figure 1, the improvements obtained from using Firth bias reduction are magnified by the backbone capacity. Is there an explanation for this?

**Summary Of The Paper:**

The submission examines the impact of Maximum-Likelihood Estimation (MLE)'s small-sample bias in the few-shot classification setting. The authors point out that while asymptotically consistent, MLE is biased in the finite-data regime, and propose to apply Firth bias reduction to few-shot classifiers to counter this bias. In this setting, the technique simplifies to a KL divergence regularizer between the uniform distribution over class labels and the learner's predictions.

Experiments on mini-ImageNet, CIFAR-FS, and tiered-ImageNet using ResNet and WideResNet network architectures show modest but consistent improvements across all settings. Firth bias reduction is shown to outperform other regularization approaches such as L2 regularization and label smoothing alternatives (confidence penalty and unigram label smoothing).


**Summary Of The Review:**

The submission is clear, well-written, and it explores a simple idea with rigor. The proposed idea is theoretically grounded and the experimental results are convincing. The one thing in the way of a higher score is that the submission could be made more accessible to readers with less experience with the theory of small-sample bias in MLE and Firth's bias reduction for MLE.

---

**Post-rebuttal update**: The authors' response addresses my concerns, and I now feel comfortable making a clear acceptance recommendation.

---

> ### Author Response · Authors · 2021-11-23
> **Response to Reviewer gyef**
>
> **Question 1:**
> The submission goes a bit quickly over the small-sample bias of MLE and Firth bias reduction for MLE. Without going into complete proofs, sketching the main arguments in both cases would make the submission accessible to a wider audience. As it stands, I feel like the submission is fully clear only to readers with prior knowledge of these notions.
>
> **Response:**
>
> We thank the reviewer for the valuable suggestion, and we agree that a simple explanation of the estimation bias before delving into complete proofs can make our work more accessible. For this, we tried to design a simple experiment in which the extent of MLE bias and the efficacy of Firth in reducing the bias can be clearly visualized. We made the first figure of our main submission to depict this simple experiment, due to its importance.
>
> To help the reader better understand the bias of MLE and how Firth penalized MLE (PMLE) removes the leading $O(N^{-1})$ term from the bias, we have added a simple simulation in which a fair coin is tossed until the first head appears (geometric distribution), and the task is to estimate the coin head probability based upon such observations. In other words, the goal is to estimate the parameter of the geometric distribution $\beta$.
>
> We chose the geometric distribution, since a closed form solution for the MLE and Firth's PMLE can be derived and can provide insightful analysis. Given the samples $(y_1, \cdots, y_N)$ from the geometric distribution, the sample mean is $\bar{y}=\frac{1}{N} \sum y_i$ and we have $\hat{\beta_{\text{MLE}}} = \frac{1}{\bar{y}}$ and $\hat{\beta_{\text{Firth}}}=\frac{N-1}{N\bar{y}- 1}$ ($\hat{\beta_{\text{Firth}}}$ can be derived by using the Taylor expansion of the MLE, which is not crucial for understanding the results here and we thus omitted the derivation). Note that the MLE is statistically consistent; that is, as $N$ increases, the MLE converges to the true generative parameter:
>
> $$\lim_{N\rightarrow \infty} \hat{\beta_{\text{MLE}}} = \lim_{N\rightarrow \infty} \frac{1}{\bar{y}} = \frac{1}{\mathbb{E}[\bar{y}]} = \beta^*.$$
>
> However, for small $N$, the $\bar{y}$ estimate can be extremely noisy, leading to an estimation bias in $\hat{\beta}_{\text{MLE}}$:
>
> $$\mathbb{E}[\hat{\beta_{\text{MLE}}}] = \mathbb{E}[\frac{1}{\bar{y}}] \neq \frac{1}{\mathbb{E}[\bar{y}]} = \beta^*.$$
>
> This effect is visualized in Figure 1 of the revision; the average MLE has a significant bias away from $\beta^*$ for small sample sizes, whereas the average Firth PMLE is almost the same as $\beta^*$ for all sample sizes. MLE and Firth PMLE become more similar as the sample size increases.
>
> To further validate that the MLE bias is indeed  of $O(N^{-1})$, we also plotted the MLE bias against the sample size in the log-log scale in the right panel of Figure 1. This clearly indicates that the MLE bias decreases with a rate of $O(N^{-1})$, and can be a major source of error due to its slow decrease rate.
>
> **Question 2:**
> Can the authors elaborate on why the number of novel classes subsampled for Figure 1 is 16, specifically?
>
> **Response:**
> This was done since there are 16 validation classes on mini-ImageNet, and initially we intended to test Firth on the hardest classification task with all the classes. Note that Firth can be applied to different classification tasks with varied number of novel classes (i.e., number of ways). We repeated the experiments of Figure 2 on mini-ImageNet for 5-way and 10-way classification tasks and included the results in Appendix Figure A9. These results suggest that Firth bias reduction is consistently effective across all the settings, and that its effectiveness is even more pronounced in harder classification tasks with higher ways, which is in line with the observation of Figure 4.
>
>
> **Question 3:**
> In Figure 1, the improvements obtained from using Firth bias reduction are magnified by the backbone capacity. Is there an explanation for this?
>
> **Response:**
> We do not have a very conclusive answer to this question, mainly because this trend is not consistently observed on different benchmarks. For example, the observation of magnified Firth improvements for higher backbone capacity in Figure 2 on mini-ImageNet does not hold for the same experiments on tiered-ImageNet in Figure A13, where the ResNet 101 improvement is closer to ResNet10 than ResNet50. However, based on all of the experiments we have done so far, while Firth is consistently effective, it seems that the specific behavior of Firth bias reduction may depend on a variety of factors, including feature distribution (because of the backbones) and classifier's complexity.

---

> > ### Comment · Reviewer_gyef · 2021-11-24
> > **Updated review**
> >
> > Thank you for the clarifications! All the issues I raised were addressed to my satisfaction, so I updated my score accordingly.

---

### Author Response · Authors · 2021-11-23
**General Response to All Reviewers**

We thank all the reviewers for their constructive feedback, attention to details, and recognizing the novelty of our work in the field of few-shot learning. We have updated the submission, in which we have included all the experiments requested/suggested by the reviewers and edited the text accordingly. The detailed responses to each comment are provided below.

We bring the attention of all the reviewers to the change in the numbering of the figures in the revised version. As suggested by Reviewer gyef, we included a new figure that sketches MLE and Firth PMLE for better understanding their different behaviors in the small-sample size regime without needing to know complete proofs, and this figure has now become the first figure of the revised submission. Therefore, all the numbering of all previous main figures has shifted by one. In response to all the reviewers, we have used the new numbering of the figures and tables in the revised submission.


**References:**
1. Firth, David. "Bias reduction of maximum likelihood estimates." Biometrika 80.1 (1993): 27-38.
2. Albert, Adelin, and John A. Anderson. "On the existence of maximum likelihood estimates in logistic regression models." Biometrika 71.1 (1984): 1-10.
3. Heinze, Georg, and Michael Schemper. "A solution to the problem of separation in logistic regression." Statistics in medicine 21.16 (2002): 2409-2419.
4. Heinze, Georg, and Michael Schemper. "A solution to the problem of monotone likelihood in Cox regression." Biometrics 57.1 (2001): 114-119.
5. Bishop, Christopher M. Pattern Recognition and Machine Learning. New York :Springer, 2006.
6. Wang, Yan and Chao, Wei-Lun and Weinberger, Kilian Q.  and van der Maaten, Laurens. "Simpleshot: Revisiting nearest-neighbor classification for few-shot learning." arXiv, 2019.
7. Yang, Shuo, Lu Liu, and Min Xu. "Free lunch for few-shot learning: Distribution calibration." arXiv preprint arXiv:2101.06395 (2021).

---

### Decision · Program_Chairs · 2022-01-20

**Decision:**

Accept (Spotlight)

**Comment:**

This work starts from the observation that maximum likelihood estimation, while consistent, has a bias on a finite sample which is likely to hurt for small sample sizes. From this, they apply Firth bias reduction to the few-shot learning setting and demonstrate its empirical benefits, notably relatively to L2 regularization or label smoothing alternatives.

After some discussion with the authors, all reviewers are supportive of this work being accepted. Two are also suggesting this work be featured as a spotlight.

The proposed method is simple, well motivated, and appears to be effective. Therefore, I'm happy to recommend this work be accepted and receive a spotlight presentation.